# Endogenous *Syngap1* alpha splice forms promote cognitive function and seizure protection

Murat Kilinc[1,2], Vineet Arora[2], Thomas K Creson[2], Camilo Rojas[2], Aliza A Le[3], Julie Lauterborn[3], Brent Wilkinson[4], Nicolas Hartel[5], Nicholas Graham[5], Adrian Reich[6], Gemma Gou[7,8], Yoichi Araki[9], Àlex Bayés[7], Marcelo Coba[4], Gary Lynch[3], Courtney A Miller[1,2], Gavin Rumbaugh[1,2]*

[1]Graduate School of Chemical and Biological Sciences, The Scripps Research Institute, Jupiter, United States; [2]Departments of Neuroscience and Molecular Medicine, The Scripps Research Institute, Jupiter, United States; [3]Department of Anatomy and Neurobiology, The University of California, Irvine, United States; [4]Zilkha Neurogenetic Institute, Keck School of Medicine, University of Southern California, Los Angeles, United States; [5]Mork Family Department of Chemical Engineering and Materials Science, University of Southern California, Los Angeles, United States; [6]Bioinformatics and Statistics Core, The Scripps Research Institute, Jupiter, United States; [7]Molecular Physiology of the Synapse Laboratory, Institut d'Investigació Biomèdica Sant Pau, Barcelona, Spain; [8]Universitat Autònoma de Barcelona, Bellaterra, Spain; [9]Department of Neuroscience, Johns Hopkins School of Medicine, Baltimore, United States

*For correspondence:
gavin@scripps.edu

**Abstract** Loss-of-function variants in *SYNGAP1* cause a developmental encephalopathy defined by cognitive impairment, autistic features, and epilepsy. *SYNGAP1* splicing leads to expression of distinct functional protein isoforms. Splicing imparts multiple cellular functions of SynGAP proteins through coding of distinct C-terminal motifs. However, it remains unknown how these different splice sequences function in vivo to regulate neuronal function and behavior. Reduced expression of SynGAP-α1/2 C-terminal splice variants in mice caused severe phenotypes, including reduced survival, impaired learning, and reduced seizure latency. In contrast, upregulation of α1/2 expression improved learning and increased seizure latency. Mice expressing α1-specific mutations, which disrupted SynGAP cellular functions without altering protein expression, promoted seizure, disrupted synapse plasticity, and impaired learning. These findings demonstrate that endogenous SynGAP isoforms with α1/2 spliced sequences promote cognitive function and impart seizure protection. Regulation of SynGAP-αexpression or function may be a viable therapeutic strategy to broadly improve cognitive function and mitigate seizure.

## Editor's evaluation

This study used three different mouse lines with altered expression of splice variants of SynGAP1 and reports that SynGAP1-α variants are more important than the SynGAP1-β variants for the regulation of cognitive function and seizure protection in mice. Given the well-known importance of the SYNGAP1 mutations in the pathophysiology of neurodevelopmental disorders, and the key regulatory roles of SynGAP1 for excitatory synaptic functions, these results provide timely and comprehensive data supporting the in vivo functions of individual SynGAP1 splice variants, including the α-1/2 variants, and suggests the therapeutic potential of increasing specific SynGAP1-α variants.

## Introduction

Pathogenic variation in *SYNGAP1*, the gene encoding SynGAP proteins, is a leading cause of sporadic neurodevelopmental disorders (NDDs) defined by impaired cognitive function, seizure, autistic features, and challenging behaviors (*Deciphering Developmental Disorders Study, 2015*; *Deciphering Developmental Disorders Study, 2017*; *Hamdan et al., 2009*; *Vlaskamp et al., 2019*; *Parker et al., 2015*; *Mignot et al., 2016*; *Iossifov et al., 2014*; *Satterstrom et al., 2020*). De novo loss-of-function variants leading to *SYNGAP1* haploinsufficiency cause a genetically defined developmental encephalopathy (ICD-10 code: F78.A1) that overlaps substantially with diagnoses of generalized epilepsy, global developmental delay, intellectual disability, and autism (*Vlaskamp et al., 2019*; *Parker et al., 2015*; *Mignot et al., 2016*; *Holder et al., 1993*; *Weldon et al., 2018*). *SYNGAP1* is completely intolerant of loss-of-function (LOF) variants (*Llamosas et al., 2020*). Thus, the presence of a clear LOF variant in a patient will lead to the diagnosis of a *SYNGAP1*-mediated developmental encephalopathy. The range of neuropsychiatric disorders causally linked to *SYNGAP1* pathogenicity, combined with the complete penetrance of LOF variants in humans, demonstrate the crucial role that this gene plays in the development and function of neural circuits that promote cognitive abilities, behavioral adaptations, and balanced excitability.

SynGAP proteins have diverse cellular functions (*Llamosas et al., 2020*; *Kilinc et al., 2018*; *Gamache et al., 2020*). The best characterized of these is the regulation of excitatory synapse structure and function located on forebrain glutamatergic projection neurons. In these synapses, SynGAP is predominately localized within the postsynaptic density (PSD), where it exists in protein complexes with synapse-associated-protein (SAP) families (*Kim et al., 1998*; *Chen et al., 1998*). Within these complexes, SynGAP proteins regulate signaling through NMDARs, where they constrain the activity of various small GTPases through non-canonical activity of a RasGAP domain (*Kilinc et al., 2018*; *Gamache et al., 2020*). This regulation of GTPase activity is required for excitatory synapse plasticity (*Ozkan et al., 2014*; *Araki et al., 2015*). Reduced expression of SynGAP in both human and rodent neurons causes enhanced excitatory synapse function during early brain development and is a process thought to impair cognitive functioning (*Llamosas et al., 2020*; *Clement et al., 2012*; *Clement et al., 2013*). SynGAP also regulates dendritic arborization. Reduced SynGAP protein expression impairs the development of dendritic arborization in neurons derived from both rodent and human tissues (*Llamosas et al., 2020*; *Aceti et al., 2015*; *Michaelson et al., 2018*), which disrupts the function and excitability of neural networks from both species. While reduced SynGAP expression enhances postsynaptic function regardless of glutamatergic projection neuron subtype, this same perturbation has an unpredictable impact on dendritic arborization, with some neurons undergoing precocious dendritic morphogenesis (*Llamosas et al., 2020*; *Aceti et al., 2015*), while others displaying stunted morphogenesis (*Michaelson et al., 2018*). This is an example of pleiotropy, where *Syngap1* gene products have unique functions depending on the neuronal subtype, or possibly within distinct subcellular compartments of the same type of neuron.

How SynGAP performs diverse cellular functions remains unclear. One potential mechanism is through alternative splicing. Indeed, the last three exons of *Syngap1* undergo alternative splicing (*Araki et al., 2020*; *Gou et al., 2020*; *McMahon et al., 2012*), which results in four distinct C-termini (a1, a2, b, g). These SynGAP C-terminal protein isoforms are expressed in both rodents and humans, and they are spatially and temporally regulated across mammalian brain development (*Araki et al., 2020*; *Gou et al., 2020*). Moreover, protein motifs present within these differentially expressed C-termini impart SynGAP with distinct cellular functions, with α-derived motifs shown to regulate postsynapse structure and function (*Rumbaugh et al., 2006*; *Vazquez et al., 2004*), while the β-derived sequences linked to in vitro dendritic morphogenesis (*Araki et al., 2020*). Absolute abundances of C-terminal isoforms are unclear, though estimates of relative expression of each have been made in adult mice (*Araki et al., 2020*), with α1 and α2 each contributing ~40%, β contributing ~15%, and γ contributing ~5%. *Syngap1* heterozygous null mice, which model the genetic impact of *SYNGAP1* haploinsufficiency in humans, express a robust endophenotype characterized by increased horizontal activity, poor learning/memory, and seizure (*Kilinc et al., 2018*; *Ozkan et al., 2014*; *Clement et al., 2012*; *Komiyama et al., 2002*; *Sullivan et al., 2020*). Currently, it remains unknown to what extent endogenous in vivo expression of alternatively spliced isoforms contribute to systems-level endophenotypes expressed in animal models.

# Results

The last three exons of *Syngap1* undergo alternative splicing (*Figure 1A*), which results in four distinct C-termini (*Figure 1B*). Exon 19 is spliced into two reading frames (e19b/e19a) (*Figure 1C*). Because e19b lacks a stop codon, coding sequences from e20 and e21 are also included in mature transcripts. This leads to expression of α1, α2, or γ C-terminal isoforms (*Figure 1C–D*). γ isoforms arise from inclusion e20, while α1 and α2 arise from the absence of e20, but inclusion of e21. e21 itself has two reading frames, with one leading to expression of α1 while the other codes for α2 (*Figure 1E*). SynGAP-β arises from splicing of e19 into the 'a' reading frame, which contains an internal stop codon (*Figure 1C*). To address how expression or function of isoforms contribute to cognitive function, behavior, and seizure latency, we created three distinct mouse lines, each with targeted modifications within the final three exons of the *Syngap1* gene. Each line expressed a unique signature with respect to C-terminal SynGAP protein variant expression or function. For example, in the *Syngap1^{td/td}* line, α isoform expression was disrupted while β forms were upregulated (*Figure 1F–G*). In contrast, *Syngap1^{β*/β*}* mice were opposite with respect to expression of α and β isoforms, with the former upregulated and the later disrupted (*Figure 1H*). Finally, the *Syngap1^{PBM/PBM}* line, which expressed point mutations that selectively disrupted an essential function of SynGAP-α1 (*Figure 1I*), was useful for determining to what extent phenotypes in the other two lines may have been driven by upregulated or downregulated isoforms.

## Reduced α1/2 C-terminal isoform expression is associated with enhanced seizure latency and cognitive impairment

We previously reported the generation of a *Syngap1* mouse line with an insertion of an IRES-TdTomato (IRES-TD) cassette within the 3'-UTR to facilitate endogenous reporting of active *Syngap1* mRNA translation in cells (*Spicer et al., 2018*). The cassette was placed within the last *Syngap1* exon (e21) between the stop codons of α1 and α2 coding sequences (*Figure 1E*; *Figure 2A*). Our prior study reported neuronal expression of fluorescent protein and normal total SynGAP (t-SynGAP) protein expression as measured by antibodies that recognize all splice forms. Due to our interest in understanding how in vivo expression of C-terminal variants impacts brain systems and behavior, we performed an in-depth characterization of behavioral phenotypes and SynGAP isoform expression in IRES-TD mice. Heterozygous (*Syngap1^{+/td}*) breeding of IRES-TD animals resulted in offspring of expected mendelian ratios (*Figure 2B*). However, while all WT (*Syngap1^{+/+}*) mice survived during the 100-day observation period, significant post-weaning death occurred in IRES-TD mice, with approximately two-thirds of homozygous mice (*Syngap1^{td/td}*) failing to survive past PND 50 (*Figure 2B*). It is well established that complete loss of t-SynGAP protein stemming from homozygous inclusion of null alleles leads to early postnatal death (*Komiyama et al., 2002*; *Kim et al., 2003*). However, ~ 50% t-SynGAP expression, like that occurring in heterozygous KO mice (*Figure 2—figure supplement 1A*), has no impact on survival (*Komiyama et al., 2002*; *Kim et al., 2003*). Given the unexpectedly poor survival of *Syngap1^{td/td}* animals, we thoroughly examined SynGAP C-terminal isoform protein expression in this line. At PND21, when all three genotypes are abundant (*Figure 2B*), t-SynGAP protein in mouse cortex homogenate was reduced in *Syngap1^{+/td}* and *Syngap1^{td/td}* mice compared to WT controls (*Figure 2C*; *Source data 1*). Reduced t-SynGAP levels appeared to be largely driven by near-complete disruption of α1/2 protein expression from the targeted allele. Reduced α isoform expression coincided with increased protein levels of β-containing C-terminal isoforms. Even with β compensation, *Syngap1^{td/td}* mice expressed only ~50% of t-SynGAP at PND21. Whole exome sequencing was carried out in each genotype. Differential gene expression (DGE) analysis revealed only a single mRNA, *Syngap1*, was abnormally expressed (*Supplementary file 1*). There was a ~ 25% reduction in mRNA levels in both *Syngap1^{+/td}* and *Syngap1^{td/td}* mice (*Figure 2—figure supplement 1B*). While the IRES-TD cassette destabilized a proportion of *Syngap1* mRNAs, the similarity in mRNA levels from both *Syngap1^{+/td}* and *Syngap1^{td/td}* samples indicated that other mechanisms must also contribute to reduced protein expression of α1/2 isoforms. Indeed, a recent study identified 3'UTR-dependent regulation of α isoform protein expression (*Yokoi et al., 2017*), suggesting that the IRES-TD cassette is also disrupting translation of these C-terminal variants. We next addressed expression of SynGAP isoforms in adulthood. In this additional experiment, only *Syngap1^{+/+}* and *Syngap1^{+/td}* mice were used because of limited survival and poor health of homozygous mice in the post-weaning period (*Figure 2B*). The general pattern of abnormal SynGAP levels persisted into adulthood, with both α isoforms reduced by ~50% compared to WT

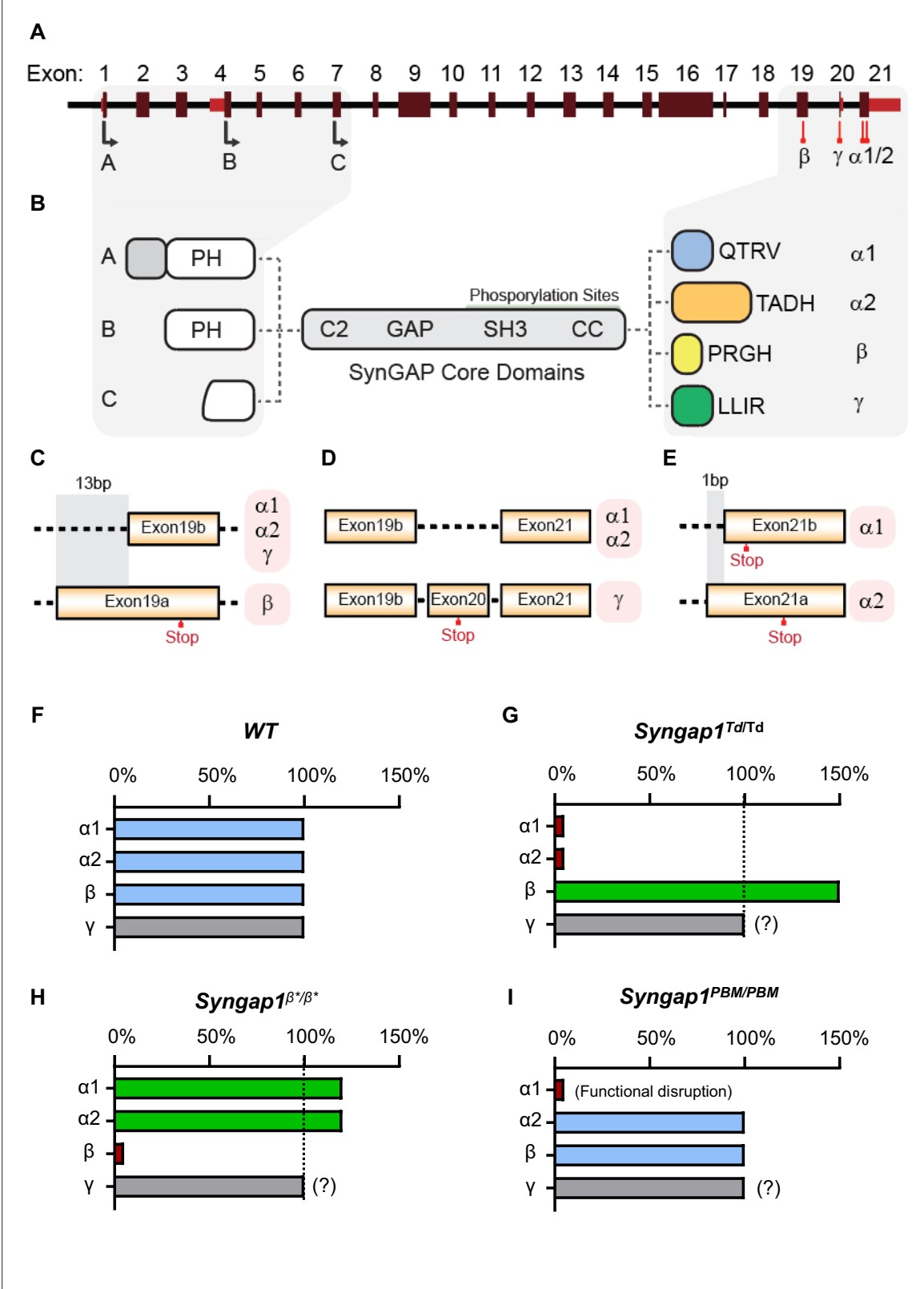

**Figure 1.** Schematic of *Syngap1 alternative splicing and summary of isoform expression in three new Syngap1 mutant mouse lines.* (**A**) Map showing alternative use of exons in N- and C-terminal isoforms. N-terminal variants are constituted via use of different start codons in exon1, 4 or 7. Exon4 is present only in B-SynGAP. C-terminal isoforms originate from use of different splice acceptors in exon 19 and 21. SynGAP-α1 contains a type-1 PDZ ligand (QTRV). Structure/function relationships of α2, β, γ isoforms remain largely unknown. (**B**) Schematics of SynGAP isoforms and protein domains. α

*Figure 1 continued on next page*

*Figure 1 continued*

and β isoforms include full Pleckstrin Homology (PH) domain. In C-SynGAP, this domain is truncated. Core regions common to all isoforms include C2, GAP (GTPase Activating Protein), Src Homology 3 (SH3)-binding, and coiled-coil (CC) domains. Multiple phosphorylation sites are present downstream of the GAP domain. (**C–E**) Schematics describing C-terminal splicing events producing distinct isoforms. (**F–I**) Summary of *Wt* and three new *Syngap1* mutant mouse lines each with distinct targeted alleles that disrupt the function or expression of SynGAP C-terminal isoforms. Bars represent expression levels of each C-terminal protein isoform relative to each *Wt* littermate control. Primary data for expression levels can be found in subsequent figures.

levels, while β isoforms were significantly enhanced (*Figure 2—figure supplement 1C*). However, the effect on t-SynGAP was less pronounced in older animals and did not rise to significance. This finding highlights the importance of measuring the expression of individual isoforms in addition to total levels of SynGAP protein in samples derived from animal or cellular models.

*Syngap1* heterozygous KO mice, which have 50% reduction of t-SynGAP and 50% reduction of all isoforms (*Figure 2—figure supplement 1—source data 1*), have normal post-weaning survival rates (*Komiyama et al., 2002*; *Kim et al., 2003*). However, survival data from *Syngap1*$^{td/td}$ mice above, which also expressed a ~ 50% reduction of t-SynGAP, but loss of α isoform expression (*Figure 2C*; *Figure 1G*), suggest that expression of these isoforms is required for survival. α isoforms are highly enriched in brain (*Araki et al., 2020*), suggesting that reduced survival stems from altered brain function. Therefore, we next sought to understand how reduced α1/2 expression (but in the context of β compensation) impacted behaviors known to be sensitive to reduced t-SynGAP expression in mice. We obtained minimal data from adult *Syngap1*$^{td/td}$ mice because they exhibit poor health and survival in the post-weaning period. However, two animals were successfully tested in the open field, and they exhibited very high levels of horizontal activity (*Figure 2D*). A more thorough characterization of behavior was carried out in adult *Syngap1*$^{+/td}$ mice, which have significantly reduced α isoforms, enhanced β expression, but relatively normal t-SynGAP levels (*Figure 2—figure supplement 1A*). *Syngap1*$^{+/td}$ mice exhibited significantly elevated open-field activity, seized more quickly in response to flurothyl, and froze less during remote contextual fear memory recall (*Figure 2E–G*). These phenotypes are all present in conventional *Syngap1*$^{+/-}$ +/- (*Ozkan et al., 2014*; *Clement et al., 2012*; *Aceti et al., 2015*; *Creson et al., 2019*), which again express ~50% reduction of all isoforms (*Figure 2—figure supplement 1A*). In contrast, Morris water maze acquisition, which is impaired in *Syngap1*$^{+/-}$ +/- (*Komiyama et al., 2002*; *Kim et al., 2003*), was unchanged in *Syngap1*$^{+/td}$ mice (*Figure 2H*). Thus, certain behaviors, including horizontal activity, freezing in response to conditioned fear, and behavioral seizure, are sensitive to reduced levels of α isoforms, but not necessarily to t-SynGAP levels. Moreover, ~ 50% loss of α1/α2 isoforms appear sufficient to disrupt long-term memory (*Figure 2G*), but insufficient to disrupt spatial learning (*Figure 2H*).

## Enhanced α1/2 C-terminal isoform expression is associated with seizure protection and improved cognitive function

The results in IRES-TD mice suggested that certain core *Syngap1*-sensitive behavioral phenotypes are caused, at least in part, by reduced α1/2 isoform expression. If α isoforms directly contribute to behavioral phenotypes in mice, then increasing their expression may drive phenotypes in the opposite direction. To test this idea, we created a new mouse line designed to upregulate SynGAP-α expression in vivo. This line, called *Syngap1*$^{β*/β*}$, contained a point mutation that prevented use of the e19a spliced reading frame (*Figure 3A–B*), the mechanism leading to expression of the SynGAP-β C-terminal variant (*Figure 1C*). This design was expected to force all mRNAs to use the e19b reading frame, leading to an increase in α variants (and loss of β expression). This line appeared healthy, bred normally, and resulting offspring were of expected Mendelian ratios. The CRISPR-engineered point mutation had the predicted impact on SynGAP isoform expression. While there was no change in t-SynGAP expression, there was a copy-number-dependent decrease in β expression, and a modest, but significant, increase in α2 expression in neonatal mice and α1 in young adult mice (*Figure 3C*; *Figure 1H*; *Figure 3—source data 1*). These animals were then evaluated in behavioral paradigms sensitive to *Syngap1* haploinsufficiency. Homozygous *Syngap1*$^{β*/β*}$ mice exhibited significantly less horizontal activity in the open field (*Figure 3D*), and also took longer to express behavioral evidence of seizure (*Figure 3E*). Further, they expressed no change in freezing levels during remote contextual memory recall (*Figure 3F*). Unexpectedly, homozygous β* mice exhibited improved learning in the Morris water maze (*Figure 3G*), with normal memory expression during the probe test. Thus, a

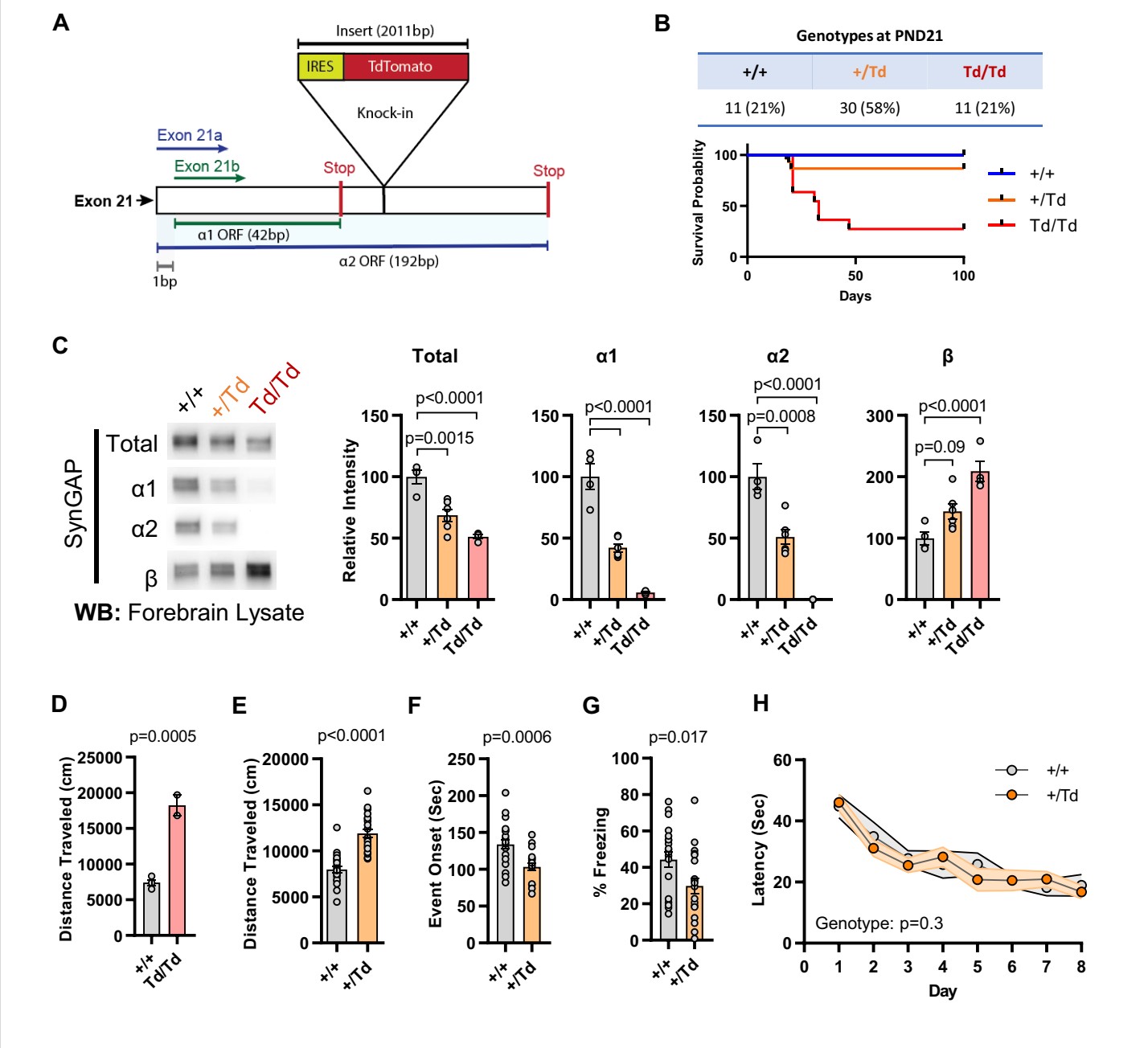

**Figure 2.** Design and characterization of *Syngap1* IRES-TdTomato knock-in mice. (**A**) IRES-TdTomato insertion site in relation to α1 and α2 open-reading frames. (**B**) Genotype ratios and survival probability following heterozygous breeding. (**C**) Representative western blots showing expression levels of total SynGAP and individual isoforms. Quantification of forebrain expression levels measured by western blot analysis. Relative intensity of bands normalized to total protein signal. Only α1 signal is significantly changed. ANOVA with Tukey's multiple comparisons test, F(2, 14) = 24.86, n = 5, p < 0.0001 (**D**) Quantification of total distance traveled in open field test in adult WT or Td/Td mice. Unpaired t-test t(4)=10.42. Note that very few homozygous Td/Td mouse survived through adulthood. (**E**) Quantification of total distance traveled in open field test in adult WT or +/Td mice. Unpaired t-test t(18)=9.007 (**F**) Latency of event onset was measured as the time taken to 1st clonus (seizure onset). Unpaired t-test: t(18)=2.588. (**G**) Percent freezing in remote contextual fear memory paradigm. Unpaired t-test: t(41)=2.49 (**H**) Plots demonstrating latency to find platform across days in Morris Water Maze training. Linear mixed model for repeated measures. n = 9–12, +/+ vs + /Td, p = 0.3.

The online version of this article includes the following source data and figure supplement(s) for figure 2:

**Source data 1.** Representative blots and total protein profiles.

**Figure supplement 1.** mRNA and protein isoform expression in Syngap1 mouse models.

**Figure supplement 1—source data 1.** Representative blots and total protein profiles.

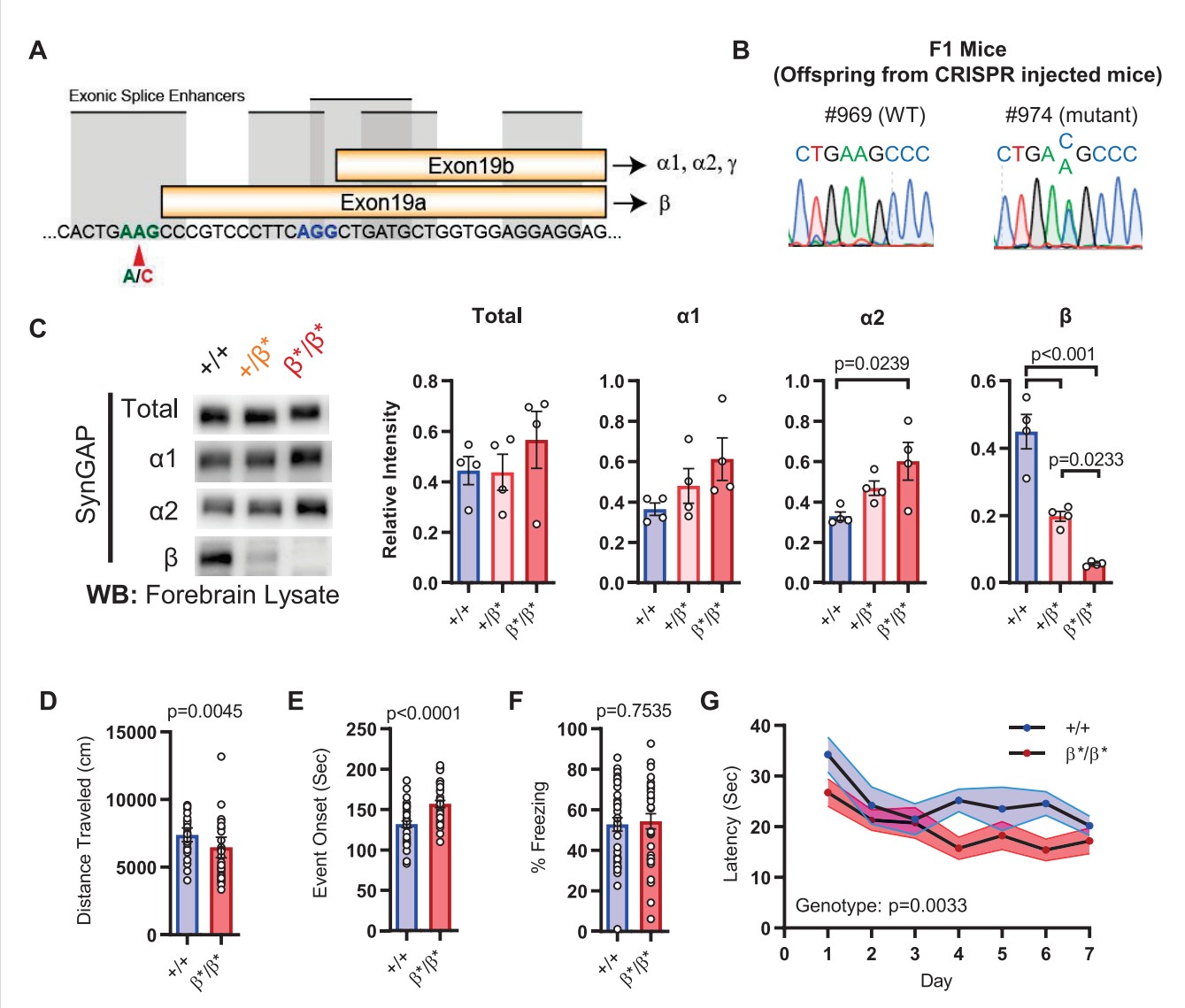

**Figure 3.** Design and characterization of *Syngap1β\** knock-in mice. (**A**) Alternative use of exon19 in distinct splicing events. Exon19 can be spliced into 2 frames shifted by 13 bp. Use of early splice acceptor (green) results in a frameshift and expresses β isoform. Use of the late splice acceptor (blue) allows expression of all other SynGAP C-terminal variants. To specifically disrupt SynGAP-β, a point mutation (A to C) was introduced to the early splice acceptor (indicated with red arrow). (**B**) Sequence trace of *Syngap1β\*/+* mice obtained via crossing F0 founders to wild-type mice. Mutation site exhibits equal levels of A and C signal in sequence trace indicating heterozygosity. (**C**) Representative western blots showing expression levels of total SynGAP and individual isoforms at PND7. Relative intensity of bands normalized to total protein signal. ANOVA with Tukey's multiple comparisons test. Total: $F_{(2, 9)} = 0.7427$, $p = 0.5029$. α1: $F_{(2, 9)} = 2.391$, $p = 0.147$. α2: $F_{(2, 9)} = 5.333$, $p = 0.0297$. β: $F_{(2, 9)} = 42.53$, $p < 000.1$ (**D**) Quantification of total distance traveled in OFT. +/+ (n = 36), β/β (n = 32); Mann-Whitney U = 346, $p = 0.0045$. (**E**) Seizure threshold was measured as the time taken to reach three separate events of 1st clonus (event onset) during the procedure. Unpaired t-test $t_{(66)}=4.237$. (**F**) Percent freezing in remote contextual fear memory paradigm. % Freezing: $t_{(66)}=0.3153$. (**G**) Plots demonstrating latency to find platform across days in Morris Water Maze training session. Statistical significance was determined by using linear mixed model for repeated measures. Genotype: $F_{(1, 15)} = 12.22$, $p = 0.0033$.

The online version of this article includes the following source data and figure supplement(s) for figure 3:

**Source data 1.** Representative blots and total protein profiles.

**Figure supplement 1.** Further characterization of Syngap1 Beta mutant mice.

**Figure supplement 1—source data 1.** Representative blots and total protein profiles.

significant increase in α isoform expression (*in the presence of nearly absent β*; *Figure 1H*) protected against seizure and improved behavioral measures associated with cognitive function, such as learning during spatial navigation.

Given the observation of seizure protection and improved learning in *Syngap1*$^{β*/β*}$ mice, we were curious if the impact of the β allele was penetrant in a *Syngap1* heterozygous (*Syngap1*$^{+/-}$) background. This is important given that *Syngap1* heterozygous mice, which model genetic impacts of *SYNGAP1* haploinsufficiency in humans, have seizures and significant cognitive impairments. To test this idea, we crossed *Syngap1*$^{+/β*}$ and *Syngap1*$^{-/+}$ +, which yielded offspring with four distinct genotypes: *Syngap1*$^{+/+}$, *Syngap1*$^{+/β*}$, *Syngap1*$^{-/+}$, *Syngap1*$^{-/β*}$ (*Figure 4A*). We first measured t-SynGAP protein in each of the four genotypes. In general terms, offspring from this cross expressed changes in SynGAP protein levels that were predicted by the known impact of each allele. For example, the effect of the *Syngap1* null allele (by comparing *Syngap1*$^{+/+}$ to *Syngap1*$^{-/+}$ samples) was to cause a significant reduction in t-SynGAP, and each of the measured C-terminal isoforms compared to *Syngap1*$^{+/+}$ (WT) animals (; *Figure 4—figure supplement 1—source data 1*). The effect of the *Syngap1*β* allele was to increase both α1 and α2 expression, and decrease β expression, whether the *Syngap1* null allele was present or absent, and these effects were also present at two developmental time points (*Figure 4B–C*, *Figure 4—figure supplement 1*). Given these results, we next performed behavioral analyses on all four genotypes. Results on behavioral endophenotypes were consistent with changes in SynGAP protein. For example, the *Syngap1* null allele impaired performance in each of the three behavioral tests performed. Comparing *Syngap1*$^{+/+}$ to *Syngap1*$^{-/+}$ animals revealed an increase in horizontal distance in the open field, faster time to seizure, and reduced freezing during remote contextual fear recall (*Figure 4D–F*; two-way ANOVA; null (-) allele, $p < 0.05$). These results replicate many past studies demonstrating the sensitivity of these behaviors to *Syngap1* haploinsufficiency in mice (*Kilinc et al., 2018*; *Clement et al., 2012*; *Aceti et al., 2015*; *Michaelson et al., 2018*; *Komiyama et al., 2002*; *Creson et al., 2019*; *Guo et al., 2009*). Interestingly, for both open field and seizure threshold tests, the presence of β* allele significantly improved measures in both WT (*Syngap1*$^{+/+}$) and *Syngap1* heterozygous (*Syngap1*$^{-/+}$) backgrounds (*Figure 4D–E*; two-way ANOVA; β* allele, $p < 0.01$; interaction of null and β alleles, $p > 0.5$). These findings were consistent with behavioral results from homozygous β* mice in the prior study (*Figure 3F–G*) and demonstrated that these two behavioral tests are sensitive to the presence of a single β* allele. Also consistent with the prior study in *Syngap1*$^{β*/β*}$ mice, the β* allele had no impact on freezing during remote contextual fear recall in either WT or *Syngap1* heterozygous backgrounds (*Figure 4F*). Thus, the β* allele partially rescued phenotypes caused by *Syngap1* heterozygosity.

## Alpha1 C-terminal isoform function is required for cognitive function and seizure protection

The results obtained from *Syngap1* IRES-TD and β* mouse lines indicated that a respective decrease, or increase, in α1/2 isoform expression impaired, or improved, behavioral phenotypes known to be sensitive to *Syngap1* heterozygosity. However, it is also possible that compensatory changes in β expression underlies these phenotypes. This alternative is unlikely, given that α and β expression is anticorrelated in both mouse lines. Thus, for β to drive phenotypes, its expression would need to be both anti-cognitive and pro-seizure, which is inconsistent with isoform expression patterns in *Syngap1*$^{-/+}$ mice (*Figure 2—figure supplement 1A*), where all protein variants are reduced by half. To directly test the hypothesis that behavioral phenotypes are sensitive to the presence of α isoforms, we attempted to create a third mouse line with point mutations that selectively impacted α isoforms, with minimal effect to SynGAP-β. We took advantage of a known molecular function exclusive to SynGAP-α1. This C-terminal variant is the only isoform that expresses a PDZ-binding motif (PBM). Importantly, cell-based studies have shown that the α1-exclusive PBM imparts unique cellular functions to this isoform (*Araki et al., 2015*; *Zeng et al., 2016*), such as the ability to become enriched at the post-synaptic density through liquid-liquid phase separation (LLPS). Past studies have shown that mutating the PBM disrupts the ability of SynGAP to regulate synapse structural and functional properties (*Rumbaugh et al., 2006*; *Vazquez et al., 2004*), including glutamatergic synapse transmission and dendritic spine size. Before this mouse could be engineered, we had to first identify PBM-disrupting point mutations within the α1 coding sequence that were silent within the open reading frames of the remaining C-terminal isoforms. In silico predictions and prior studies (*Rumbaugh et al.,*

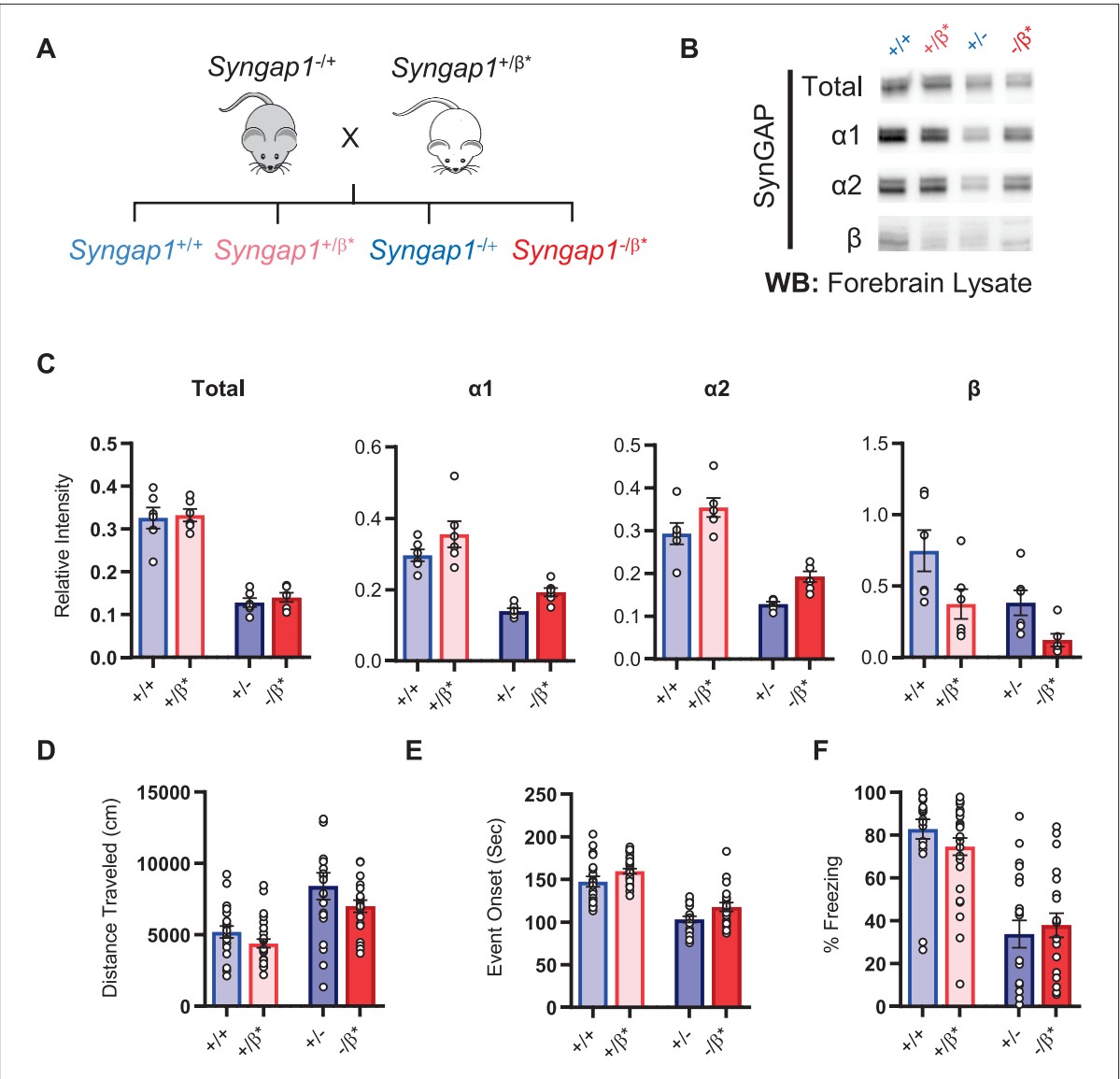

**Figure 4.** Characterization of offspring derived from *Syngap1*<sup>+/-</sup> *and Syngap1*<sup>β\*/+</sup> cross-breeding. (**A**) Breeding scheme for offspring genotypes for Syngap1 <sup>+/-</sup>and *Syngap1*<sup>+/β\*</sup> lines. (**B**) Representative western blots showing expression levels of total SynGAP and individual isoforms at PND7 for all genotypes. (**C**) Quantification of B. Two-way ANOVA with Tukey's multiple comparison test. **Total:** (-) allele $F_{(1, 20)} = 146.3$, $p < 0.0001$; β\* allele $F_{(1, 20)} = 0.3344$, $p = 0.5696$. Allelic Interaction $F_{(1, 20)} = 0.03191$, $p = 0.8600$. **α1:** (-) allele $F_{(1, 20)} = 56.01$, $p < 0.0001$; β\* allele $F_{(1, 20)} = 7.009$, $p = 0.0155$; Allelic Interaction $F_{(1, 20)} = 0.02397$, $p = 0.8785$. **α2:** (-) allele $F_{(1, 20)} = 81.79$, $p < 0.0001$; β\* allele $F_{(1, 20)} = 11.92$, $p = 0.0025$; Allelic Interaction $F_{(1, 20)} = 0.0044$, $p = 0.9479$. **β:** (-) allele $F_{(1, 20)} = 9.149$, $p = 0.0067$; β\* allele $F_{(1, 20)} = 9.676$, $p = 0.0055$; Allelic Interaction $F_{(1, 20)} = 0.3027$, $p = 0.5883$. (**D**) Quantification of total distance traveled in open field test. Two-way ANOVA with Tukey's multiple comparison test. (-) allele $F_{(1, 86)} = 28.85$, $p < 0.0001$; β\* allele $F_{(1, 86)} = 4.132$, $p = 0.0452$; Allelic Interaction $F_{(1, 86)} = 0.2951$, $p = 0.5884$ (**E**) Latency of event onset was measured as the time taken to 1st clonus (seizure onset). Two-way ANOVA with Tukey's multiple comparison test. (-) allele $F_{(1, 82)} = 91.71$, $p < 0.0001$; β\* allele $F_{(1, 82)} = 8.967$, $p = 0.0036$; Allelic Interaction $F_{(1, 82)} = 0.07333$, $p = 0.7872$ (**F**) Percent freezing in remote contextual fear memory paradigm. Two-way ANOVA with Tukey's multiple comparison test. (-) allele $F_{(1, 86)} = 69.37$, $p < 0.0001$; β\* allele $F_{(1, 86)} = 0.1544$, $p = 0.6953$; Allelic Interaction $F_{(1, 86)} = 1.392$, $p = 0.2414$.

The online version of this article includes the following source data and figure supplement(s) for figure 4:

**Source data 1.** Representative blots and total protein profiles.

**Figure supplement 1.** Representative western blots showing expression levels of total SynGAP and individual isoforms at PND60 for all genotypes.

**Figure supplement 1—source data 1.** Representative blots and total protein profiles.

2006; Zeng et al., 2016) suggested that a double point mutation within the α1 PBM could meet these requirements (Figure 5A–B). To test this prediction, we introduced these point mutations into a cDNA that encoded the PBM and then tested how this impacted PDZ binding. Using an established cell-based assay that reports PDZ binding between the SynGAP PBM and PSD95 (Zeng et al., 2016), we found that these point mutations had a large effect on SynGAP-PDZ binding. When expressed individually in HeLa cells, PSD95-tRFP localized to the cytoplasm, while a SynGAP fragment containing the coiled-coil domain and α1 C-tail (EGFP-CCα1) was enriched in the nucleus (Figure 5C–E). The co-expression of these two proteins led to SynGAP localization into the cytoplasm. However, this shift in localization did not occur when PBM point mutations were present (Figure 5D–E), indicating that the selected amino acid substitutions severely impaired binding to the PDZ domains. Moreover, co-immunoprecipitation in heterologous cells indicated that the point mutations in the PBM disrupted the direct association of full-length SynGAP-α1 with PSD95 (Figure 5—figure supplement 1—source data 1). Finally, these point mutations also reduced synaptic enrichment of exogenously expressed SynGAP-α1 fragments in cultured forebrain neurons (Figure 5—figure supplement 1C-E).

Based on this evidence, we introduced the PBM-disrupting point mutations into the final exon of the mouse Syngap1 gene through homologous recombination (Figure 5A and F–H). Both heterozygous and homozygous PBM mutant animals (hereafter Syngap1$^{+/PBM}$ or Syngap1$^{PBM/PBM}$) were viable, appeared healthy, and had no obvious dysmorphic features. We observed Mendelian ratios after inter-breeding Syngap1$^{+/PBM}$ animals (Figure 5—figure supplement 1F), demonstrating that disrupting the PBM had no impact on survival. Western blot analysis of forebrain homogenates isolated from Syngap1$^{+/PBM}$ or Syngap1$^{PBM/PBM}$ mutant animals demonstrated no difference in t-SynGAP protein levels using antibodies that detect all SynGAP splice variants (Figure 5—source data 1I-J). Moreover, using isoform-selective antibodies (Araki et al., 2020; Gou et al., 2019), we observed normal expression of SynGAP-β and SynGAP-α2 isoforms (Figure 5I–J). A reduced signal of ~60% was observed in samples probed with α1-specific antibodies. However, we also observed a similarly reduced signal in heterologous cells expressing a cDNA encoding the mutant PBM (Figure 5—figure supplement 1—source data 1), indicating that these antibodies have reduced affinity for the mutated α1 motif. Together, these data strongly suggest that the α1 variant is expressed normally in Syngap1$^{PBM/PBM}$ animals. This interpretation was supported by RNA-seq data, where normal levels of mRNA containing the α1 reading frame were observed in brain samples (Figure 5—figure supplement 1J). These data, combined with the observation of no change in total SynGAP protein expression in Syngap1$^{PBM/PBM}$ samples (Figure 5I–J), strongly support the conclusion that the PBM-disrupting point mutations do not change the expression levels of the major SynGAP C-terminal splice variants, including those containing the PBM. Thus, this animal model is suitable for understanding the putative biological functions mediated by α1-specific splicing.

Given the disruption to SynGAP-α1 PBM, we sought to understand how altering this functional motif impacted previously defined features of SynGAP at excitatory postsynapses. α1 is believed to be anchored within the PSD in part through PBM binding to PDZ domain containing proteins. However, SynGAP molecules multimerize in vivo and it is currently unknown to what extent this feature contributes to homo- vs. hetero-multimerization. Thus, it is unclear how a functional disruption to one isoform generally impacts native SynGAP complexes at synapses. This is important given that C-terminal isoforms have distinct functions within excitatory neurons (Araki et al., 2020). t-SynGAP levels were reduced in unstimulated PSD fractions prepared from either adult hippocampal homogenates or primary neurons from Syngap1$^{PBM/PBM}$ mice (). Importantly, a corresponding increase in t-SynGAP was observed in the triton-soluble synaptosomal fraction, further supporting the observation of reduced t-SynGAP levels in the PSD. PSD abundance of SynGAP-β and α2 isoforms were not significantly different in PBM mice compared to WT littermates (Figure 6A), which suggested that α1 may exist in distinct biochemical complexes compared to other C-terminal isoforms (i.e. homomeric SynGAP-α1 complexes). Unfortunately, this could not be tested directly in these samples due to reduced affinity of α1-specific antibodies in Syngap1$^{PBM/PBM}$ mice (Figure 5—figure supplement 1H). Therefore, we performed an additional experiment to address the potential existence of isoform-specific biochemical complexes. This required culturing neurons, inducing chemical LTP (cLTP), and then measuring how the stimulus impacted PSD abundance of total SynGAP and C-terminal isoforms. First, we found that a typical cLTP paradigm drove extrusion of total SynGAP from the PSD of WT mice (Figure 6—figure supplement 1—source data 1), while a weak cLTP stimulation did not (Figure 6—figure

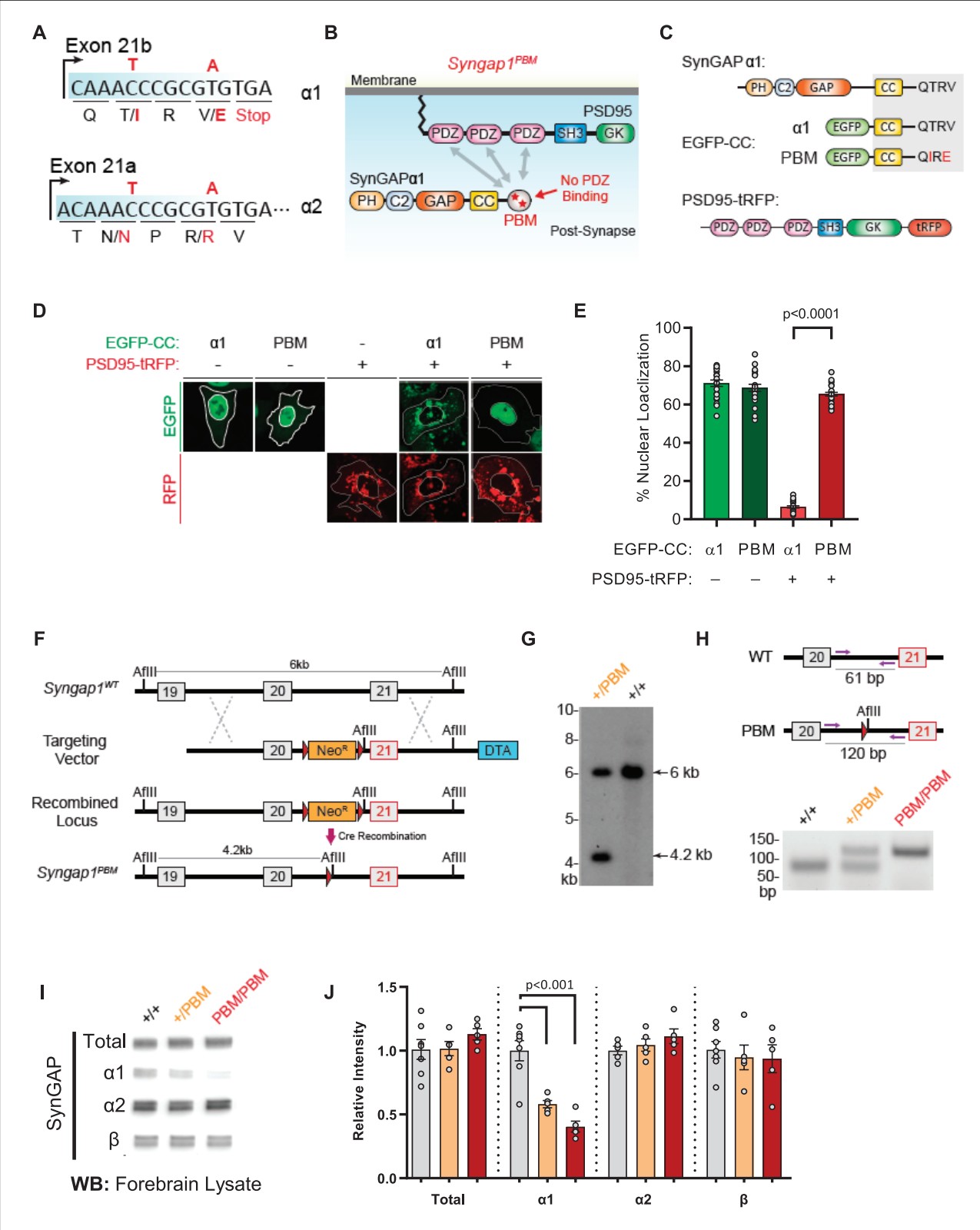

**Figure 5.** Validation of SynGAP PDZ binding motif (PBM) mutations and construction of the *Syngap1^PBM* mouse line. (**A**) Schematic diagram for exon map and alternative use of Exon21 in *Syngap1* gene. Exon21b encodes for α1 isoform. Exon 21 a encodes for α2 isoform. Point mutations indicated in red alter exon 21b coding sequence without influencing exon21a open reading frame. (**B**) Schematics of SynGAPα1 and PSD95 domain structure and the location of point mutations. (**C**) Illustrations of constructs expressed in HeLa cells to study PDZ-dependent interaction between SynGAP and PSD95.

*Figure 5 continued*

EGFP-CC constructs are homologous to SynGAPα1 C-terminus. (**D**) Co-localization of EGFP-CCα1 and PSD95-tRFP in HeLa Cells. Representative images showing subcellular localizations of WT or PDZ-binding mutant (PBM) EGFP-CCα1 and PSD95-tRFP in HeLa cells when expressed individually or together. (**E**) Quantification of (**D**). Nuclear localization is calculated as the ratio of EGFP signal colocalized with DAPI vs total EGFP intensity in within an individual cell. ANOVA with Tukey's multiple comparisons test, $F_{(3, 96)} = 531.4$. $p < 0.0001$ (**F**) Schematics of the targeting strategy. The targeting vector was spanning Exon20 and 21. The vector included point mutations in Exon21, a neomycin resistance selection cassette flanked by Cre recombination sites and diphtheria toxin selection cassette (DTA). (**G**) Southern blot analysis showing the genomic DNA of the tested heterozygous mice compared to C57BL/6 J wild-type DNA. The AflII digested DNAs were blotted on nylon membrane and hybridized with external 5' probe spanning exon19. (**H**) PCR based genotyping strategy. Primers flanking leftover LoxP site yields 61 bp product in WT and 120 bp product in mutated allele. (**I**) Representative western blots showing expression levels of total SynGAP and individual isoforms in forebrain lysates. (**J**) Quantification of I. Relative intensity of bands normalized to total protein signal. Only α1 signal is significantly changed. ANOVA with Tukey's multiple comparisons test, $F_{(2, 14)} = 24.86$, n = 5.

The online version of this article includes the following source data and figure supplement(s) for figure 5:

**Source data 1.** Representative blots.

**Figure supplement 1.** Determining impact of proposed PBM coding mutations on SynGAP protein.

**Figure supplement 1—source data 1.** Representative blots.

supplement 1B), which are findings consistent with past studies that defined SynGAP dynamics within biochemical fractions or subcellular compartments (*Araki et al., 2015*; *Araki et al., 2020*). In contrast, weak cLTP was capable of driving a reduction in total SynGAP from PSDs in *Syngap1^PBM/PBM^* mice (*Figure 6—figure supplement 1B*). Immunoblotting with isoform-specific antibodies in the weak cLTP condition provided insight into the differential behavior of total SynGAP in PSDs from WT vs. PBM mice. For example, weak cLTP was sufficient to drive reduced PSD abundance for both α2 and β isoforms in both WT and PBM neurons (*Figure 6—figure supplement 1B*). However, α1 PSD abundance was unchanged in WT mice after weak cLTP, demonstrating that this isoform, when intact, exhibits distinct properties in response to synaptic NMDAR activation. Replicating this approach in PBM neurons revealed that this distinct feature of α1 was due to the existence of an intact PBM motif (*Figure 6—figure supplement 1B*). These data indicate that reduced PSD abundance of t-SynGAP in *Syngap1^PBM/PBM^* mice is driven by altered biochemical features and dynamics of α1. The other isoforms appear minimally impacted by the PBM mutation. Based on this model, spontaneous activity within *Syngap1^PBM/PBM^* neurons may drive reduced SynGAP PSD abundance, presumably by reducing the stimulus threshold required to drive α1 out of this compartment. To test this, we measured SynGAP PSD abundance and ERK1/2 signaling in WT and PBM neurons with and without activity blockers (*Figure 6B*). Acute blockade of synaptic activity normalized SynGAP levels in the PSD and ERK1/2 signaling (*Figure 6B*). Similar treatments also normalized enrichment of SynGAP in dendritic spines and surface expression of GluA1 in neurons derived from *Syngap1^PBM/PBM^* mice (*Figure 6C and D*). These results indicate that endogenous PBM binding of the α1 isoform regulates an activity-dependent process within excitatory synapses.

Blocking synaptic activity in *Syngap1^PBM/PBM^* neurons prevented alterations in SynGAP levels at postsynapses (*Figure 6A–D*). This suggested that the PBM regulates SynGAP-specific functions in excitatory synapses, such as activity-dependent extrusion of α1 from the PSD. However, SynGAP-α1 undergoes LLPS and this mechanism is thought to facilitate the organization of the PSD (*Zeng et al., 2016*). Thus, disrupted SynGAP post-synaptic levels could also be attributable to altered structural organization of the PSD. To determine if the PBM contributes to the organization of macromolecular complexes within excitatory synapses, we immunoprecipitated PSD95 from neurons obtained from either WT or *Syngap1^PBM/PBM^* mutant neurons. These neurons were treated with APV to avoid the confounds of elevated NMDAR signaling. These samples were then analyzed by mass spectrometry to determine how disrupting SynGAP-PDZ binding impacted the composition of PSD95 macromolecular complexes. In general, we found only minor differences in the abundance of proteins that comprise PSD95 complexes when comparing samples from each genotype (*Figure 7—source data 1A*). Only 1 out of ~161 proteins (from 133 distinct genes) known to be present within PSD95 complexes (*Li et al., 2017*) met our threshold for significance, although there were modest changes in proteins with structurally homologous PBMs (Type-1 PDZ ligands), such as Iqseq2 and Dlgap3 (*Figure 7B*). However, the vast majority of related PBM-containing proteins were not different in mutant neurons, including NMDAR subunits and TARPs (*Figure 7C*). Consistent with the mass spectrometry analysis, immunoblot analyses found no changes in TARPs or LRRTM2 in isolated PSDs from *Syngap1^PBM/PBM^* mice

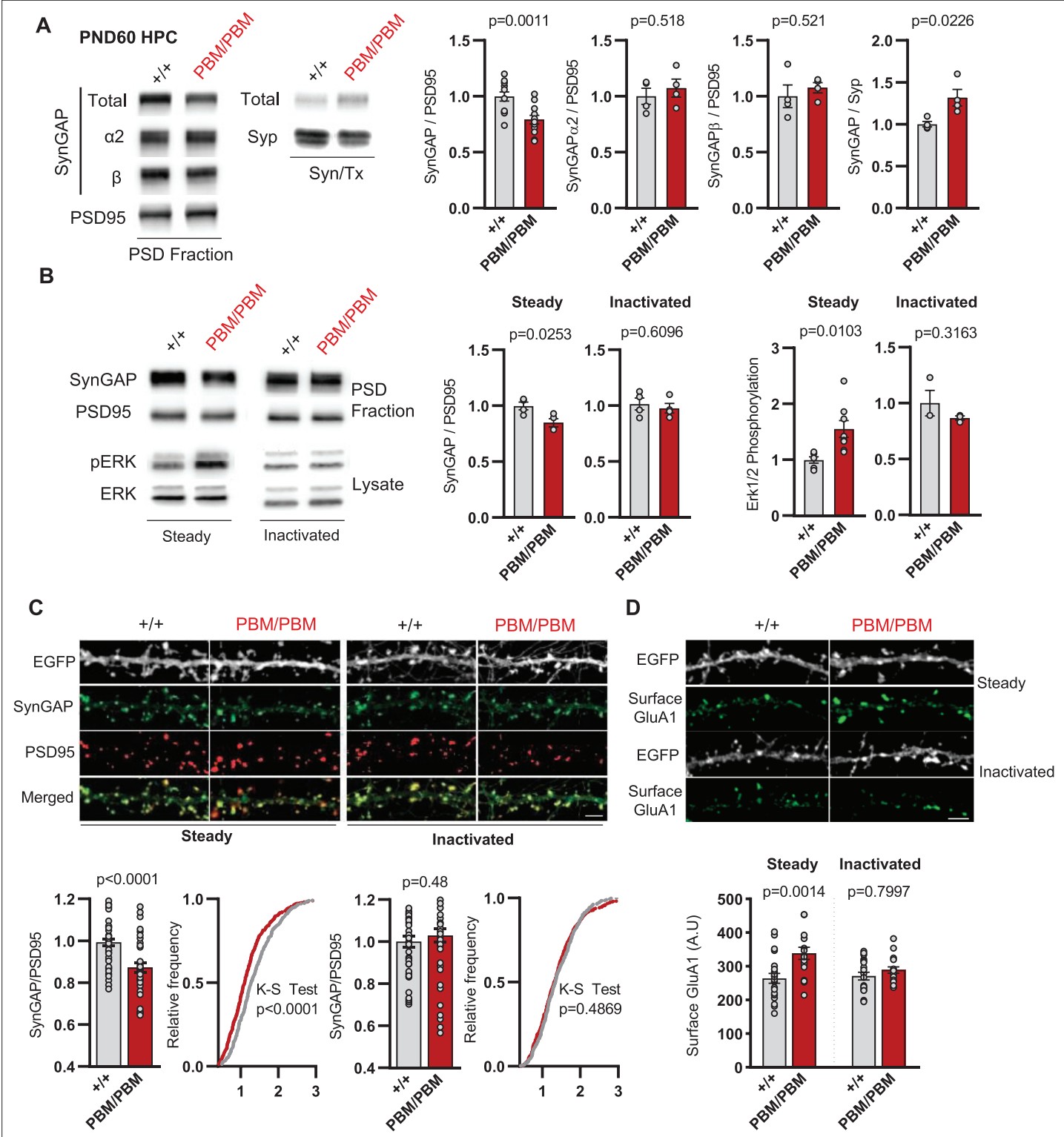

**Figure 6.** SynGAP synapse localization in *Syngap1^PBM* mouse line. (**A**) Western blots showing relative distribution of SynGAP in PSD and Syn/Tx fractions from adult hippocampi. Quantification of western blots probing SynGAP (total, α2, β), Synaptophysin and PSD95. For PSD fractions PSD95 and for Syn/Tx fractions Synaptophysin (Syp) were used as loading control. PSD fractions, Total SynGAP t(22)=3.733, p = 0.0011 n = 12 (3 technical replicates for each sample), SynGAPα2 t(6)=0.6855, p = 0.518, SynGAPβ t(6)=0.6813, p = 0.521. Syn/TX fractions Total SynGAP: t(6)=3.049, p = 0.0226, n = 4. Each sample represents hippocampi pooled from 2 mice. (**B**) Western blots showing relative enrichment of (**i**) SynGAP and PSD95 in PSD fractions isolated from DIV18-21 cultures, (ii) phospho and total-ERK1/2 levels in whole cell lysates in steady or inactivated state. Synaptic enrichment of SynGAP in (**i**) steady-

*Figure 6 continued on next page*

Figure 6 continued

state: Unpaired t-test, t(12)=3.040 p = 0.0103. (ii) inactivated state: Unpaired t-test, t(6)=0.5385 p = 0.6096. Erk1/2 phosphorylation is calculated as ratio of phospho- Erk1/2 to total-Erk1/2 in homogenates. Erk1/2 phosphorylation in (**i**) steady-state: Unpaired t-test, t(6)=2.961 p = 0.0253. (ii) inactivated state: Unpaired t-test, t(4)=1.144 p = 0.3163(**C**) Synaptic enrichment of total SynGAP in WT and PBM mutants in steady or inactivated state. Levels of SynGAP relative to PSD95 signal in dendritic spines. Left, bar graphs demonstrate mean enrichment in an individual dendritic segment. Steady-state: t(90)=4.393 p < 0.0001. Inactivated: t(78)=0.6982 p = 0.48. Cumulative distribution of SynGAP to PSD95 ratios in individual synapses. Kolmogorov-Smirnov test, Steady-state: p < 0.0001, Inactivated: p = 0.4869. (**D**) Surface GluA1 expression in primary forebrain cultures in steady or inactivated state. Quantification of mean surface GluA1 levels coincident with PSD95 puncta. Two-way ANOVA with Tukey's multiple comparisons test. Interaction: F(1,74)=4.112, p = 0.0462, Genotype: F(1,74)=11.09, p = 0.0014. Treatment: F(1,74)=2.329, p = 0.1313. Each n represents an average of 25–30 spines from a dendritic segment belonging to distinct neurons.

The online version of this article includes the following source data and figure supplement(s) for figure 6:

**Source data 1.** Representative blots.

**Figure supplement 1.** Isoform-specific regulation of SynGAP protein within the PSD.

**Figure supplement 1—source data 1.** Representative blots.

(*Figure 7—source data 2*). Although PDZ binding was disrupted, SynGAP protein levels were also unchanged within PSD95 complexes, a result consistent with PSD and synapse localization measurements in APV-treated neurons derived from *Syngap1*^PBM/PBM mice (*Figure 6B–C*). These results indicate that SynGAP interacts with PSD95 in a non-PDZ-dependent manner. In support of this interpretation, there is significant overlap between the interactomes of PSD95 (*Li et al., 2017*) and SynGAP (*Wilkinson et al., 2017*) macromolecular complexes (*Figure 7H*). Thus, within intact postsynapses, SynGAP and PSD95 interact, as part of a macromolecular complex, through binding to common protein intermediaries. Together, these data suggest that SynGAP PBM binding to PDZ domains is not a major factor promoting the organization of PSD95 macromolecular complexes or the PSD. Rather, the PBM appears to regulate SynGAP-specific mechanisms that control signaling through NMDARs.

Given that altering the SynGAP PBM disrupts signaling through NMDARs, we hypothesized that hippocampal CA1 LTP would be disrupted in *Syngap1*^PBM/PBM mice. The within-train facilitation of responses across the seven theta bursts used to induce LTP did not differ between genotypes (*Figure 8A*), indicating that standard measures of induction, including NMDAR channel activation, were not impacted by PBM mutations. However, short-term plasticity (STP; *Figure 8C and D*) and LTP (*Figure 8B and E*) were both reduced in *Syngap1*^PBM/PBM mice. The ratio of LTP/STP was no different between genotypes (*Figure 8F*). Blocking NMDAR channel function is known to disrupt both STP and LTP (*Volianskis et al., 2013*). However, a key measure of NMDA channel function was normal in PBM mutant mice (*Figure 8A*). Thus, these data are consistent with the idea that disrupting SynGAP-PDZ binding impairs signaling normally induced downstream of synaptic NMDAR activation. Synaptic plasticity, such as LTP, is thought to contribute importantly to multiple forms of learning and memory. As such, we next measured performance of WT and *Syngap1*^PBM/PBM mice in a variety of learning and memory paradigms that have previously shown sensitivity in *Syngap1* mouse models, including IRES-TD and β* lines. Behavioral analysis in this line revealed a significant increase in horizontal locomotion in the open-field test (*Figure 8G*), a significantly reduced seizure threshold (*Figure 8H*), and significantly reduced freezing during retrieval of a remote contextual fear memory (*Figure 8I*). Moreover, we also observed impaired acquisition during Morris water maze learning (*Figure 8J*). Together, these behavioral data indicate that the PBM within SynGAP-α1 splice forms is critical for learning and memory, as well as protecting against seizure.

## Alpha1/2 C-terminal isoform expression or function predicts changes in excitatory synapse function

Behavioral results from IRES-TD and PBM mice were consistent with each other, and also consistent with a reduction in all SynGAP isoforms occurring in *Syngap1* conventional heterozygous null mice. These three mouse lines share a common molecular feature – reduced expression or function of SynGAP-α1 isoforms (*Figure 1F–I*; *Supplementary file 1*). Prior studies have shown that exogenously expressed SynGAP-α1 is a negative regulator of excitatory synapse function (*Rumbaugh et al., 2006*; *Wang et al., 2013*). Thus, we hypothesized that IRES-TD and PBM mouse lines would express elevated excitatory synapse function, while *Syngap1*^β*/β* mice, which have enhanced α1 expression, would express reduced synapse function. To test this idea, we performed whole-cell voltage clamp

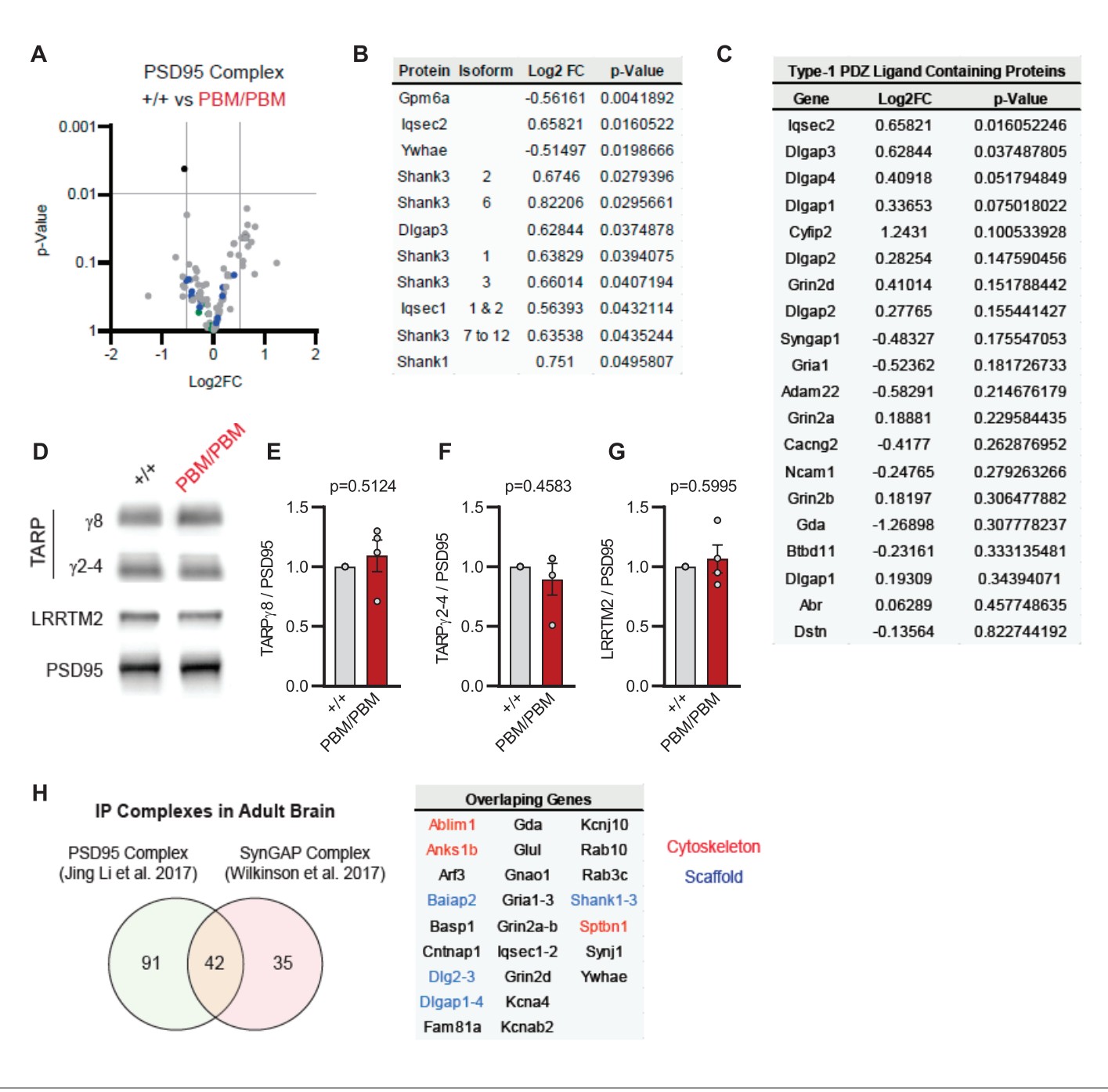

**Figure 7.** Characterization of native PSD95 complexes from *Syngap1^PBM* animals. (**A**) Volcano plot demonstrating the label-free quantitative mass-spectrometry profile of the logarithmic difference in protein levels in the immunoprecipitated PSD95 complexes derived from DIV21 +/+ and PBM/PBM cultures in inactivated state. Only Gpm6a (shown in black) was significantly altered beyond p > 0.001 cutoff. Blue dots represent proteins with type 1 PDZ-ligands. Green dots represent DLG family proteins. p Values were calculated via t-test for each protein. Samples were derived from individual cultures (4 per genotype) which are immunoprecipitated separately. Log2FC was calculated as ratio of PBM/PBM over +/+. (**B**) List of proteins that are differentially expressed beyond p > 0.05 cutoff. Note that Iqseq2 and Dlgap3 are PDZ-binding proteins. (**C**) Mass-spectrometry profile of type-1 PDZ binding motif containing proteins in immunoprecipitated PSD95 complex in +/+ vs PBM/PBM inactivated cultures. (**D**) Western blots showing relative expression of TARPs and Lrrtm2 in PSD fractions from adult hippocampi in +/+ vs PBM/PBM. (**E–G**) Quantifications of (**D**). (**E**) TARPg8 t(6)=0.6961, *P* = 0.5124. (**F**) TARPg2-4 t(6)=0.7924, p = 0.4583 (**G**) Lrrtm2 t(6)=0.5542, p = 0.5995. Each sample represents hippocampi pooled from 2 mice. (**H**) Comparison of PSD95 and SynGAP IP complexes as reported by *Li et al., 2017* and *Wilkinson et al., 2017*. Note that PSD95 and SynGAP complexes share diverse range of components involving cytoskeletal and scaffolding proteins.

*Figure 7 continued on next page*

*Figure 7 continued*

The online version of this article includes the following source data for figure 7:

**Source data 1.** Mass Spec raw data.

**Source data 2.** Representative blots.

recordings in acute somatosensory cortex slices derived from all three of these lines because these neurons have been shown to be sensitive to *Syngap1* heterozygosity in ex vivo slice preparations (*Michaelson et al., 2018*). PBM mice exhibited a modest increase in *m*EPSCs amplitude and a more substantial increase in *m*EPSC frequency, two measures consistent with enhanced postsynaptic function (*Figure 9A–C*). We also observed increased excitatory synapse function (both *m*EPSC amplitude and frequency distributions) in IRES-TD mice (*Figure 9D–F*). The sample size for PBM mEPSC analysis is somewhat underpowered, although these significant effects agree with independent mEPSC observations from the IRES-TD mice. Moreover, effects on *m*EPSC amplitude in L2/3 SSC neurons observed in both lines are similar to what has been reported previously in *Syngap1*$^{+/-}$ +/- (*Michaelson et al., 2018*). In contrast, *Syngap1*$^{\beta*/\beta*}$ mice, which have significantly elevated α1/α2 expression, expressed reduced *m*EPSC amplitude and frequency measurements relative to littermate control slices (*Figure 9G–I*), a phenotype consistent with SynGAP-α1 overexpression in excitatory neurons (*Rumbaugh et al., 2006*; *Wang et al., 2013*).

## Discussion

In this study, we created three distinct mouse lines, each regulating the expression or function of SynGAP protein isoforms (*Figure 1F–I*), without appreciable change in total SynGAP expression levels. A summary of all measured phenotypes in these lines can be found in *Supplementary file 1*. The overall conclusion from this study is that α-containing SynGAP isoforms promote cognitive functions that support learning/memory, while also protecting against seizure. It is important to understand the relationship between SynGAP isoform function and systems-level manifestations of the different isoforms, such as behavioral expression related to cognitive function and seizure. It has been shown previously that *Syngap1* C-terminal splicing imparts distinct cellular functions of SynGAP proteins (*Araki et al., 2020*; *McMahon et al., 2012*; *Rumbaugh et al., 2006*; *Vazquez et al., 2004*). Thus, targeting endogenous isoform expression in animal models presents an opportunity to determine to what extent distinct cellular functions of SynGAP could contribute to various intermediate phenotypes present in *Syngap1* mouse models. Given that *SYNGAP1* is a well-established NDD gene and LOF mutations are highly penetrant in the human population (*Deciphering Developmental Disorders Study, 2015*; *Deciphering Developmental Disorders Study, 2017*; *Hamdan et al., 2009*; *Parker et al., 2015*; *Mignot et al., 2016*; *Satterstrom et al., 2020*; *Hamdan et al., 2011*; *Berryer et al., 2013*), studying these relationships have the potential to provide much needed insight into the neurobiology underlying human cognitive and behavioral disorders that first manifest during development. Second, there is increasing interest in targeted treatments for patients with *SYNGAP1* disorders due to the penetrance of LOF variants, the relatively homogenous manifestations of the disorder (e.g. cognitive impairment and epilepsy), and the growing number of patients identified with this disorder (*Lim et al., 2020*). Restoring SynGAP protein expression in brain cells is the most logical targeted treatment for this disorder because most known patients have de novo variants that cause genetic haploinsufficiency (*Holder et al., 1993*). The most logical therapeutic approach would be to reactivate native expression of the endogenous gene. However, the findings from this study indicate that targeted therapies for *SYNGAP1* disorders that enhance expression of α isoforms may be sufficient to provide a benefit to patients. Indeed, only a modest upregulation of α1/2 expression within a *Syngap1* heterozygous background was sufficient to improve behavioral deficits commonly observed in that mouse line (*Figure 4*). Third, the discovery that SynGAP-α1/2 expression/function is pro-cognitive and provides protection from seizure suggests that these isoforms, and the cellular mechanisms that they regulate, could be harnessed to intervene in idiopathic cognitive and excitability disorders, such as neurodegenerative disorders and/or epilepsies with unknown etiology.

Several lines of evidence from this study support the conclusion that SynGAP-α isoform expression or function promotes cognition and seizure protection. IRES-TD and PBM mouse lines each had similar learning/memory and seizure threshold phenotypes, with both mouse lines exhibiting impaired

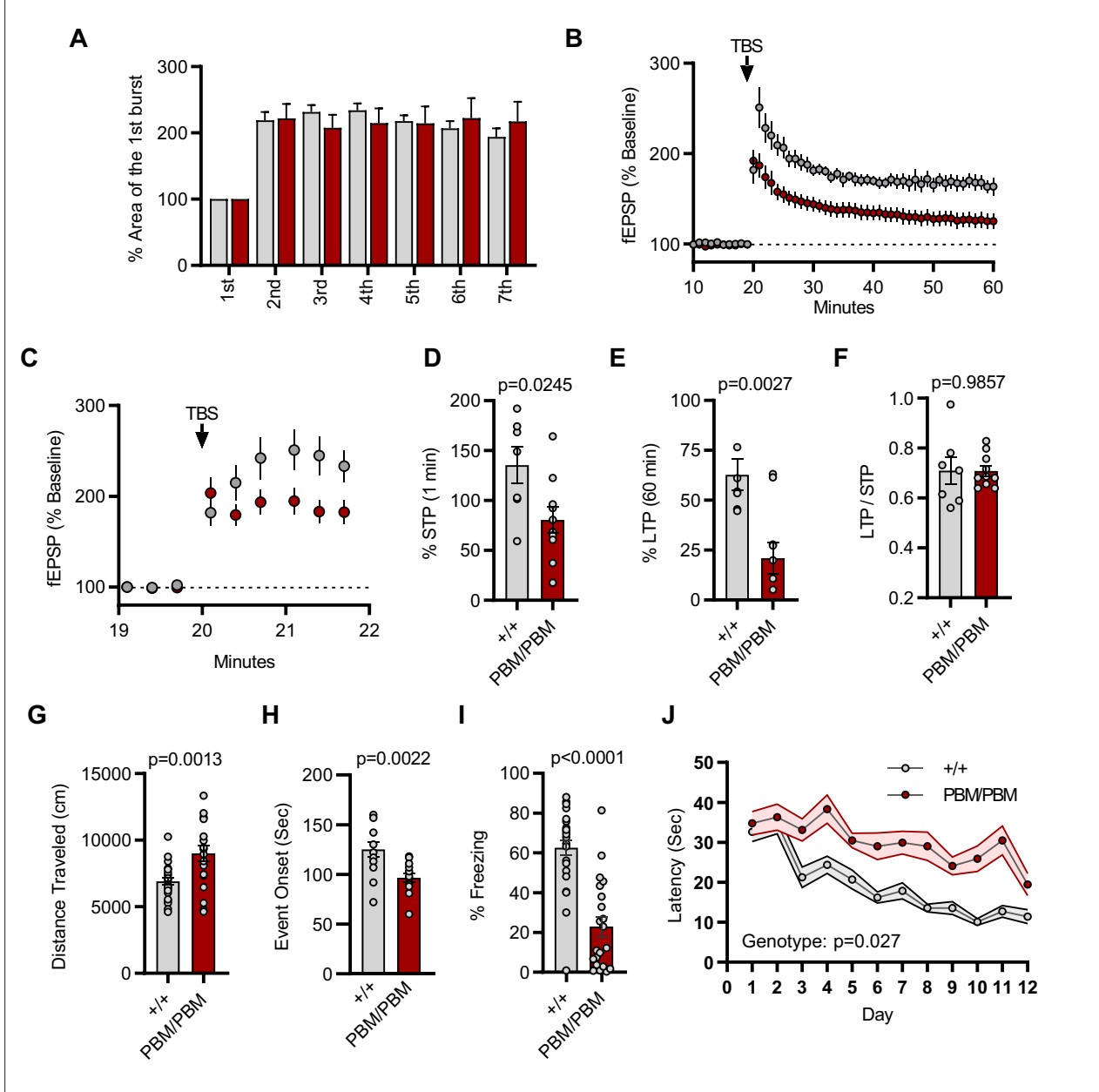

**Figure 8.** Plasticity and behavior deficits in the *Syngap1^PBM* mouse line. (**A**) Facilitation of burst responses was calculated by expressing the area of the composite fEPSP corresponding to the 2nd theta burst within each train as a fraction of the 1st burst response. No statistically significant difference was found between genotypes (wildtypes are shown in gray and PBM/PBM are in red). (**B**) Magnitude of long-term potentiation (LTP) following delivery of a single train of five theta bursts. The slope of the fEPSP was normalized to the mean value for a 20 min baseline period; shown are group means and standard errors. (**C**) Percent fEPSP during and immediately after the LTP induction. Note that homozygous mutants reach to peak potential immediately following TBS. (**D**) Bar graph shows % potentiation in 1 min after stimulus. t(15)=2.499, p = 0.0245. (**E**) Bar graph shows % potentiation in 60 min after stimulus. t(15)=3.594, p = 0.0027. (**F**) LTP to STP ratio of individual slices. Note that the level of LTP is proportional to the degree of acute potentiation (1 min after stimulus). t(15)=0.01818, p = 0.9857. (**G**) Quantification of total distance traveled in OFT. t(45)=3.427, p = 0.0013. (**H**) Seizure threshold was measured as the time taken to reach three separate events of 1st clonus (event onset) during the procedure. Unpaired t-test t(25)=3.420 p = 0.0022. (**I**) Percent freezing in remote contextual fear memory paradigm. % Freezing: t(45)=6.463, p < 0.0001. (**J**) Plots demonstrating latency to find platform across days in Morris Water Maze training session. Statistical significance was determined by using linear mixed model for repeated measures. n = 14, +/+ vs PBM/PBM, p = 0.027.

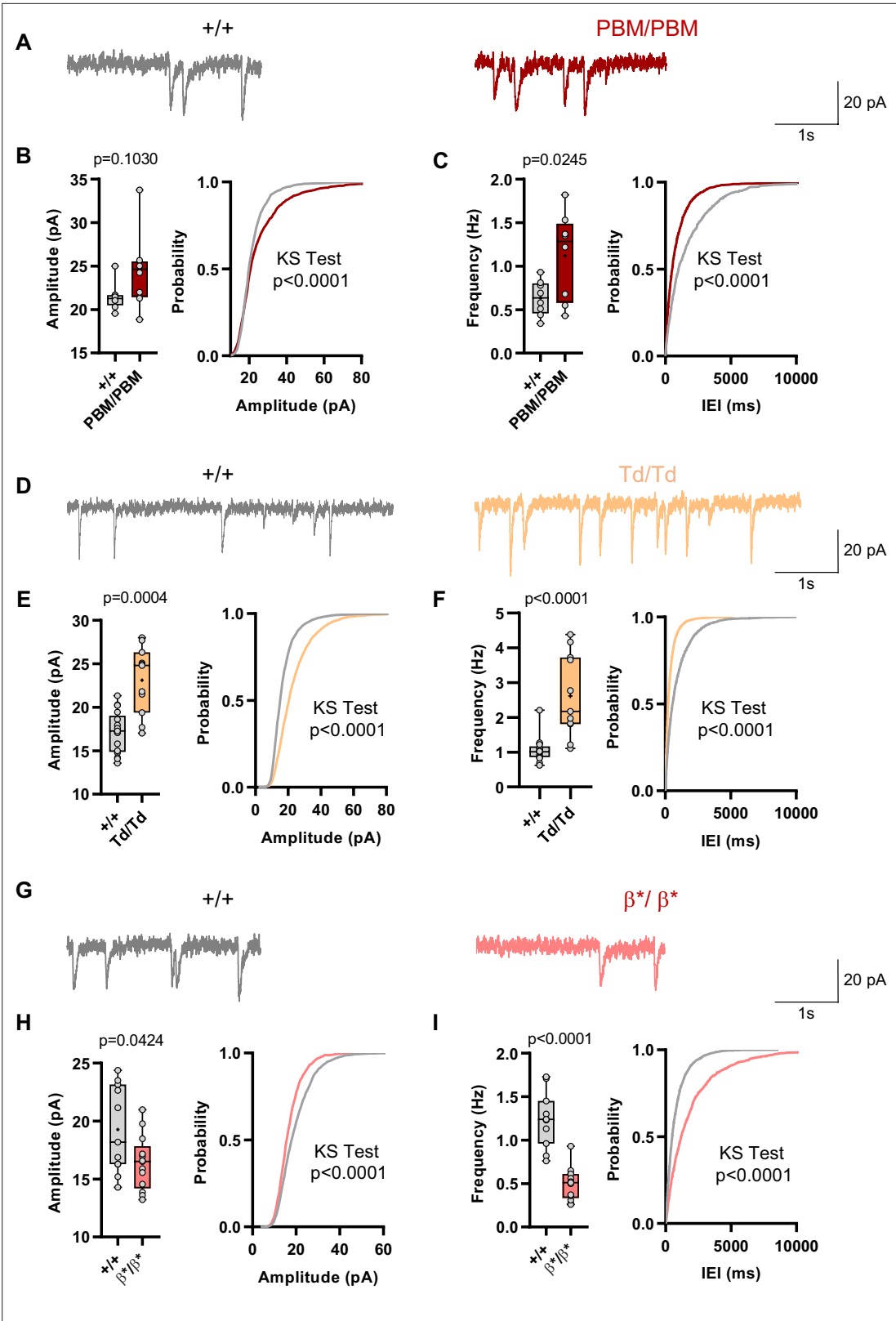

**Figure 9.** Analysis of excitatory synapse function in *Syngap1*<sup>PBM</sup>, *Syngap1*<sup>β\*</sup>, and *Syngap1*<sup>td</sup> mouse lines. (**A**) Representative mEPSCs traces from L2/3 SSC in +/+ vs PBM/PBM (**B**) Scatter plots and cumulative histograms showing trend towards increase but no significant difference in Amplitudes of mEPSCs +/+ vs PBM/PBM (**C**) Scatter plots and cumulative histograms showing significant increase in frequency of mEPSCs +/+ vs PBM/PBM. Unpaired t test: p = 0.0245, n = 8 for each genotype. (**D**) Representative mEPSCs traces from L2/3 SSC in +/+ vs Td/Td. (**E**) Scatter plots and cumulative histograms

*Figure 9 continued on next page*

*Figure 9 continued*

showing significantly increased amplitudes of mEPSCs in +/+ vs Td/Td. Unpaired t test: p = 0.0004, n = 17 cells for +/+, n = 11 cells for Td/Td mice. (**F**) Scatter plots and cumulative histograms showing significant increase in frequency of mEPSCs in +/+ vs Td/Td. Unpaired t test: p < 0.0001, n = 17 cells for +/+, n = 11 cells for Td/Td mice. (**G**) Representative mEPSCs traces from L2/3 SSC in +/+ vs β*/β*. (**H**) Scatter plots and cumulative histograms showing significantly decreased amplitudes of mEPSCs in L2/3 SSC for +/+ vs β*/β*. Unpaired t test: p = 0.0424, n = 11 cells for +/+, n = 13 cells for β*/β*. (**I**) Scatter plots and cumulative histograms showing significant decrease in frequency of mEPSCs in +/+ vs β*/β*. Unpaired t test: p < 0.0001, n = 11 cells for +/+, n = 13 cells for β*/β*.

phenotypes related to these two types of behavioral analyses. Indeed, these two mouse lines also shared a common molecular perturbation - reduced expression or function of alpha isoform(s). For example, IRES-TD homozygous mice lacked expression of both α1 and α2 isoforms and these animals exhibited severe phenotypes, including reduced post-weaning survival and dramatically elevated horizontal activity in the open field. Additional phenotypes were also present in heterozygous IRES-TD mice, which underwent more comprehensive testing because of better survival in the post-weaning period. These additional phenotypes included reduced seizure threshold and impaired freezing during a remote contextual fear expression test. PBM homozygous mice had normal expression of SynGAP protein, but lacked a functional domain present exclusively in α1 isoforms, a type-1 PDZ-binding domain. PBM homozygous mice shared phenotypes with IRES-TD mice, including impaired remote contextual fear expression, elevated horizontal activity in the open field, and a reduced seizure threshold. These mice also expressed impaired learning during Morris water maze acquisition. Importantly, these behavioral phenotypes are well established in *Syngap1* heterozygous mice (*Ozkan et al., 2014*; *Clement et al., 2012*; *Aceti et al., 2015*; *Creson et al., 2019*; *Guo et al., 2009*), indicating that SynGAP protein loss-of-function underlies these abnormalities. Thus, it reasonable to speculate that α isoform LOF is one potential mechanism underlying these behavioral abnormalities. Dysregulation of excitatory synapse function in cortical circuits is one of many possible cellular mechanisms underlying common phenotypes in IRES-TD and PBM mutant mice lines. Whole cell electrophysiology experiments from developing cortical neurons in situ from each line revealed evidence of elevated excitatory synapse strength during the known *Syngap1* mouse critical period. Indeed, elevated excitatory synapse strength in developing forebrain glutamatergic neurons is a major cellular outcome present in *Syngap1* heterozygous knockout mice (*Ozkan et al., 2014*; *Clement et al., 2012*; *Clement et al., 2013*; *Michaelson et al., 2018*). Moreover, elevated excitatory synapse strength is consistent with impaired cognitive function and reduced seizure threshold.

Studies in the *Syngap1*β* line also support this interpretation. These mice were devoid of SynGAP-β protein expression, yet we did not observe cellular or behavioral phenotypes consistent with *Syngap1* heterozygosity. Rather surprisingly, mice lacking SynGAP-β expression had intermediate phenotypes that opposed what was commonly observed in *Syngap1* heterozygous KO mice (and shared by IRES-TD/PBM lines). For example, β* mice exhibited improved spatial learning in the Morris water maze, reduced horizontal activity in the open field, and an elevated seizure threshold (evidence of seizure protection). These phenotypes were modest in effect size, but highly significant. These phenotypes were reproducible because open field and seizure phenotypes were also present in a separate series of experiments performed in the *Syngap1* heterozygous background. This demonstrates that the impact of the β* allele is penetrant even when expression of isoforms is reduced by half compared to WT mice. As a result, the β* allele partially rescued open field and seizure phenotypes present in *Syngap1*$^{+/-}$ +/-. For impaired β expression to drive phenotypes, expression of this isoform would be anticorrelated with cognitive function and seizure protection. Put another way, reduced β expression would need to enhance phenotypes and increased expression of these isoforms would need to disrupt them. This outcome is unlikely given that it is inconsistent with phenotypes observed in *Syngap1*$^{+/-}$ +/-, which have reduced expression of all isoforms, including SynGAP-β.

Phenotypes in β* mice are likely driven by significantly elevated SynGAP-α expression rather than reduced SynGAP-β. Electrophysiological studies in these mice revealed reduced excitatory neuron synaptic strength, a finding consistent with exogenously elevated SynGAP-α1 expression (*Rumbaugh et al., 2006*; *Wang et al., 2013*). Moreover, these synapse-level results are consistent with seizure protection observed in β* mice. Phenotypes in PBM mice also support this hypothesis. This model does not have altered t-SynGAP expression, or a change in β expression. Yet, the behavioral- and synapse-level phenotypes are consistent with those observed in IRES-TD and *Syngap1*$^{+/-}$ +/-. The

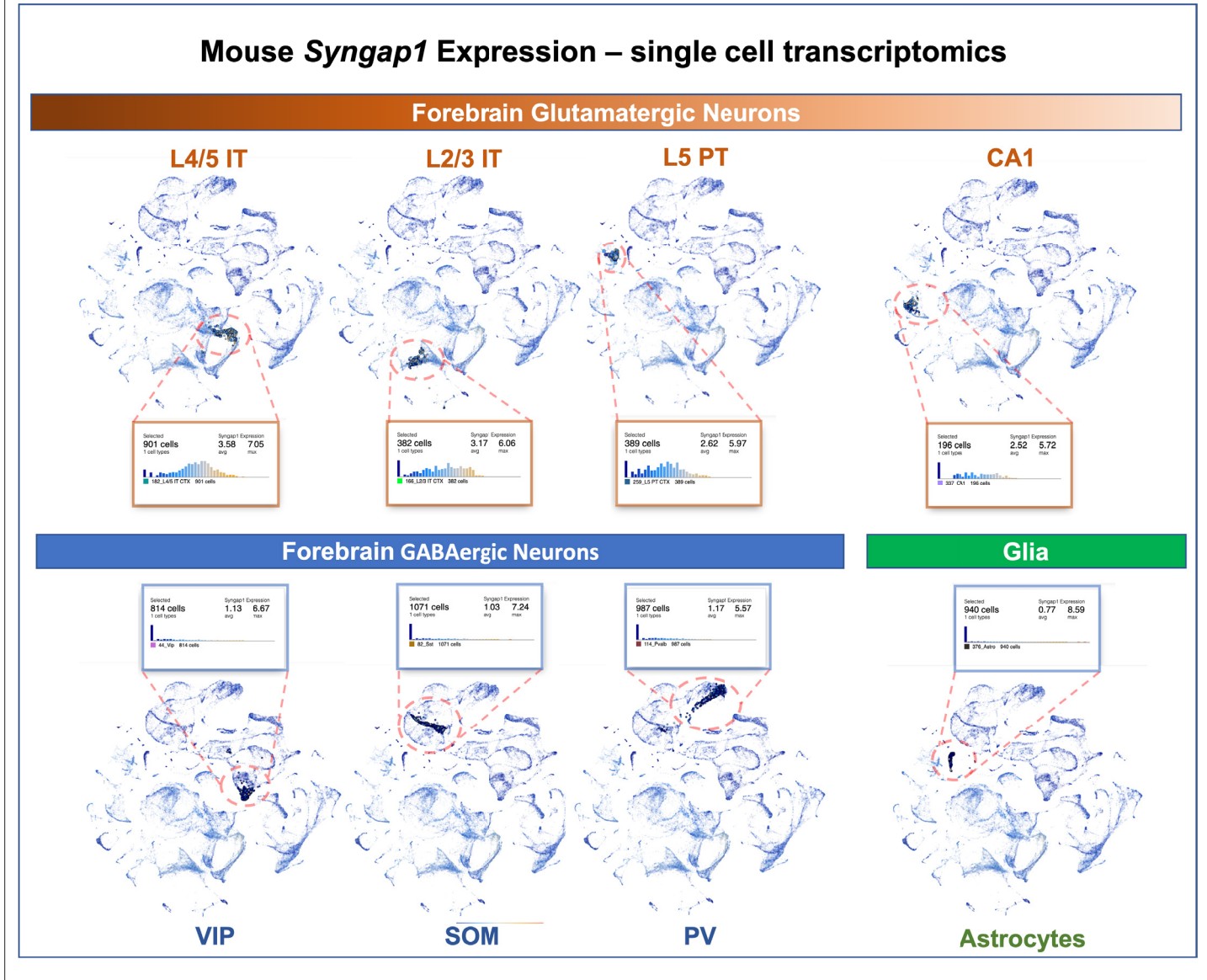

**Figure 10.** Single-cell mRNA expression of *Syngap1* in mouse cortex and hippocampus. Allen Brain Map single cell transcriptomics database was mined and summarize to note expression data for *Syngap1* in representative cell types in mouse cortex and hippocampus. Original data can be found using the following URL - https://celltypes.brain-map.org/rnaseq/mouse_ctx-hpf_smart-seq?selectedVisualization=Scatter+Plot&colorByFeature=Gene+Expression&colorByFeatureValue=Syngap1.

observation that α isoforms promote cognitive function and seizure protection are consistent with known molecular functions of these isoforms, at least with respect to regulation of synapse strength and resultant impacts on neural circuit function. For example, α1 imparts SynGAP with the ability to undergo liquid-liquid phase transitions (*Zeng et al., 2016*). This biophysical process is associated with regulation of Ras signaling in dendritic spines required for AMPA receptor trafficking that supports use-dependent synapse plasticity (*Araki et al., 2015*; *Araki et al., 2020*). Input-specific plasticity is crucial during development to sculpt the assembly of neural circuits (*Zhang and Poo, 2001*), while also being important in mature circuits to promote experience-dependent changes in already-established circuitry (*Lynch et al., 2007*).

*Syngap1* is a potent regulator of forebrain glutamatergic neuron biology and that many phenotypes observed in models of *Syngap1* regulation have origins in these excitatory neurons. Single cell transcriptomics data from adult mice indicate that *Syngap1* is principally expressed in glutamatergic neurons in the cortex and hippocampus rather than GABAergic interneurons (*Figure 10*).

Single-cell mRNA expression data agree with experimental evidence of SynGAP protein expression in rodent neurons. For example, SynGAP protein expression is enriched in glutamatergic neurons (**Kim et al., 1998**; **Rumbaugh et al., 2006**; **Kim et al., 2003**), with relatively high levels in upper lamina of isocortex (**Butko et al., 2013**). Other studies show that SynGAP protein is absent from several types of forebrain GABAergic neurons, but was expressed in a subpopulation of morphologically distinct inhibitory cells (**Zhang et al., 1999**). Expression data agree with prior experimental observations from electrophysiological and behavioral measurements in mice where *Syngap1* expression was conditionally regulated in distinct neuronal subtypes. Commonly observed and robust phenotypes observed in *Syngap1* heterozygous null mice (**Guo et al., 2009**; **Muhia et al., 2010**) were phenocopied in animals where *Syngap1* heterozygosity was restricted to excitatory neurons in the forebrain (**Ozkan et al., 2014**). Moreover, major electrophysiological and behavioral phenotypes in *Syngap1* heterozygous mice were also rescued when *gene* expression was restored in in this same population (**Ozkan et al., 2014**). In contrast, only minor phenotypes emerged in mice when *Syngap1* expression was disrupted in GABAergic neurons. A separate group reported similar results. In that study, most *Syngap1* heterozygous mouse behavioral phenotypes were insensitive to selective disruption within a GABAergic neuron population, although one behavioral measure of cognition was mildly affected (**Berryer et al., 2016**).

Alpha isoforms, and α1 in particular, exhibit enrichment in dendritic spine synapses (**Gou et al., 2020**). As a result, baseline synaptic phenotypes related to *Syngap1* gene expression appear dominated by the ability of both α1 and α2 isoforms to suppress excitatory synapse function. Studies from several research groups have shown that SynGAP-α1 is a negative regulator of excitatory synapse structure and function (**Araki et al., 2015**; **Araki et al., 2020**; **Rumbaugh et al., 2006**; **Vazquez et al., 2004**; **Wang et al., 2013**). In contrast, the role of α2 isoform protein function on excitatory synapse structure/function is less clear. One study suggested that α2 has an opposing function relative to α1 within excitatory synapses, with the former acting as an enhancer, rather than a suppresser, of excitatory synapse function (**McMahon et al., 2012**). However, a more recent study demonstrated that α2 has a similar, albeit less robust ability to suppress AMPA receptor content within dendritic spines (**Araki et al., 2020**), indicating that it too can act as a negative regulator of synapse function. Our results here support the view that both α1 and α2 can act as suppressors of excitatory synapse function. In our studies, α1 and α2 were both co-regulated in the IRES-TD and β* lines, with both isoforms downregulated in the former and upregulated in the latter. In both mouse lines, baseline excitatory synapse strength was inversely proportional to expression levels of α1/2 isoforms. If α1 and α2 had opposing functions at the synapse level, then co-regulation of both isoforms would be expected to lead to no significant differences in synapse function. As a result, we hypothesize that improvements in spatial learning and protection from seizure in β* mice arise through changes in excitatory synapse biology mediated by α isoforms. Thus, in-depth study of α isoform biology at both the cell biological and neural systems levels may reveal molecular and cellular approaches to improve cognition and mitigate uncontrolled excitability.

It is important to note that our interpretation that β* mouse phenotypes are most likely driven by changes in αisoforms does not preclude a fundamental role of β in sculpting neural systems, or that reduced expression of this isoform in *Syngap1*[+/-] +/- has no role in disease pathobiology. Rather, our results highlight the importance of endogenous α isoforms in regulating excitatory synapse function and associated behavioral outcomes. What is known about the function of other C-terminal protein variants, such as β and γ? A recent study suggested that β and γ isoforms lack the ability to regulate excitatory synapse function, further strengthening the idea that α isoforms account for *Syngap1*-dependent regulation of excitatory synapse function (**Araki et al., 2020**). However, *Syngap1* is known to regulate additional cellular process beyond regulation of excitatory synapse function, such as dendritic morphogenesis and patterning in vivo (**Clement et al., 2012**; **Aceti et al., 2015**; **Michaelson et al., 2018**). Evidence suggests that all isoforms can regulate dendritic morphogenesis in vitro, although SynGAP-β was shown to be a stronger regulator of this process relative to the other C-terminal isoforms (**Araki et al., 2020**). In vivo, β was found to be expressed earlier in development and to be less enriched in the postsynaptic density compared to other variants (**Gou et al., 2020**). Thus, β is well positioned to regulate non-synapse related neuronal processes. Future studies will be required to elucidate the specific cellular functions of non-alpha isoforms and how they contribute to the development of neural function and behavior. Given the complexities of *Syngap1* regulation on

dendritic morphogenesis (*Aceti et al., 2015*; *Michaelson et al., 2018*), and the direct linkage between dendritic morphogenesis and circuit function in cortex in *Syngap1* mutant animals (*Michaelson et al., 2018*), future studies on the function of individual isoforms would ideally be carried out in vivo in developing animals.

# Materials and methods

## Key resources table

| Reagent type (species) or resource | Designation | Source or reference | Identifiers | Additional information |
|---|---|---|---|---|
| Biological sample (*Mus musculus*) | Mouse primary forebrain neurons | This study | | 18–21 days in vitro |
| Biological sample (*Mus musculus*) | Cortical and hippocampal tissue | Multiple strains (this study) | | Male and female |
| Strain, strain background (*Mus musculus*) | IRES-TD | *Spicer et al., 2018* | | C57BL/6 J background |
| Strain, strain background (*Mus musculus*) | Beta KO (β*) | This study | | C57BL/6 J background |
| Strain, strain background (*Mus musculus*) | PBM | This study | | C57BL/6 J background |
| Cell line (*Homo sapiens*) | HeLa | Farzan Lab (Scripps Research) | | |
| Cell line (*Homo sapiens*) | H293T | Kissil Lab (Scripps Research) | | |
| Genetic reagent (*Rattus norvegicus*) | PSD95-tRFP | Addgene | #52,671 | Plasmid |
| Genetic reagent (*Mus musculus*) | EGFP-CCα1 | This study | | Plasmid |
| Genetic reagent (*Mus musculus*) | EGFP-CCPBM | This study | | Plasmid |
| Genetic reagent (*Mus musculus*) | EGFP-SynGAPα1 | This study | | Plasmid |
| Genetic reagent (*Mus musculus*) | EGFP-SynGAPα1PBM | This study | | Plasmid |
| Antibody | Rabbit polyclonal antibody | Thermo | PA1-046 | Anti-total SynGAP (1:1000) |
| Antibody | Rabbit polyclonal antibody | Millipore | 06–900 | Anti-SynGAPα1 (1:1000) |
| Antibody | Rabbit monoclonal antibody | Abcam | ab77235 | Anti-SynGAPα2 (1:1000) |
| Antibody | Rabbit polyclonal antibody | *Araki et al., 2020* Huganir Lab (JHU) | | Anti-SynGAPβ (1:1000) |
| Antibody | Mouse monoclonal antibody | Thermo | MA1-045 | Anti-PSD95 (1:2000) |
| Antibody | Rabbit polyclonal antibody | Novus | NB300-653 | Anti-Synaptophysin (1:1000) |
| Antibody | Mouse monoclonal antibody | CST | 9,106 | Anti-phosphoERK (1:1000) |
| Antibody | Mouse monoclonal antibody | CST | 4,696 | Anti-ERK (1:1000) |
| Antibody | Mouse monoclonal antibody | Millipore | MAB2263 | Anti-GluA1 N-term. (1:500) |
| Antibody | Rabbit polyclonal antibody | Millipore | Ab9876 | Anti-TARP (1:500) |
| Antibody | Rabbit polyclonal antibody | Thermo Pierce | PA521097 | Anti-LRRTM2 (1:1000) |
| Sequence-based reagent | IRES-TD Genotyping Primer Fw | IDT | | AGATCCACCAGGCCCTGAA |
| Sequence-based reagent | IRES-TD Genotyping Primer Rev | IDT | | GTCTTGAACTCCACCAGGTAGTG |
| Sequence-based reagent | PBM Genotyping Primer Fw | IDT | | CTGGTTCAAAGGCTCCTGGTA |
| Sequence-based reagent | PBM Genotyping Primer Rev | IDT | | CTGTTTGTTTCTCACCTCCAGGAA |

*Continued on next page*

*Continued*

| Reagent type (species) or resource | Designation | Source or reference | Identifiers | Additional information |
|---|---|---|---|---|
| Other | CamKII.Cre | Addgene | 105558-AAV9 | *Adeno-associated virus* (AAV) |
| Other | CAG.Flex.EGFP | Addgene | 28304-PHPeB | *Adeno-associated virus* (AAV) |
| Commercial assay or kit | Pierce BCA Protein Assay Kit | Pierce | 23,225 | |
| Chemical compound, drug | D-AP5 | Tocris | 0106 | |
| Chemical compound, drug | Bicuculline | Tocris | 0109 | |
| Chemical compound, drug | Tetrodotoxin | Tocris | 1,069 | |
| Chemical compound, drug | Glycine | Tocris | 0219 | |
| Chemical compound, drug | Strychnine | Sigma | S7001-25G | |
| Software, algorithm | Prism 8 | Graphpad | | |
| Software, algorithm | ImageJ(Fiji) | NIH | | |

## Animals

This study was performed in strict accordance with the recommendations in the Guide for the Care and Use of Laboratory Animals of the National Institutes of Health. All the animals were handled according to approved institutional animal care and use committee (IACUC) protocols of The Scripps Research Institute.

$Syngap1^{PBM}$ and $Syngap1^{Td}$ mice were constructed in collaboration with genOway (France). The targeting vector was electroporated into ES cells derived from the inner cell mass of 3.5 days old C57BL/6 N embryos. Cells were then subjected to negative and/or positive selection(s) before the presence of the correct recombination event was validated by PCR and Southern blot. ES cell clones with verified mutations were injected into blastocysts which were implanted into pseudo-pregnant females to obtain chimeras. Chimeric mice were bred with C57BL/6 Cre-deleter mice to excise the Neomycin selection cassette and to generate heterozygous mice carrying the Neo-excised knock-in allele. Progeny were genotyped by PCR. The recombinase-mediated excision event was further validated by Southern blot using 5' external probes. Knock-in lines were maintained on C57BL/6 J background and bred for 3 generations prior to experimental use. $Syngap1^{PBM}$ animals were genotyped using the following primers, which amplified the locus spanning the LoxP site: Fwd: 5'-ctggttcaaaggc tcctggta-3' Rev: 5'- ctgtttgtttctcacctccaggaa-3'. This combination yielded a 61 bp product in WT and 120 bp product in knock-in alleles. $Syngap1^{Td}$ line were genotyped using the primers amplifying the locus including the TdTomato cassette: Fwd: 5'-AGATCCACCAGGCCCTGAA-3' Rev: 5'- GTCTTGA ACTCCACCAGGTAGTG-3'.

$Syngap1$-β* mice were constructed in collaboration with the Scripps Research Genetics core facility. To selectively disrupt SynGAP-β expression, exon19a splice acceptor site 'AAG' was mutated into 'ACG'. To introduce the point mutation, purified CRISPR/Cas9 protein combined with gRNA and donor DNA was injected to ~100 zygotes and implanted into surrogate mice. A 200 bp PAGE purified ss-oligo repair template centering the CRISPR cut site was used as donor DNA. Recombination events were detected by PCR and Sanger sequencing of the DNA isolated from tails of F0 potential founders. This process identified 2 chimeric mice with evidence of the targeted nucleotide variants. Chimeras were then bred with C57BL6/J and resultant heterozygous F1 mice were used to start the colony. Because CRISPR carries a risk of off-target genomic effects, prior to any downstream experiments, this line was further crossed into C57BL6/J for >3 generations.

## Transcriptomics

PND7 mice forebrains (Cortex + hippocampus) were immediately removed and stored in RNALater (Thermo, AM7020). mRNA was isolated with RNeasy mini kit (74104, Qiagen). RNA integrity was measured using Agilent 2,100 Bioanalyzer (RIN value ≥ 9.2 for each sample). Library preparation and sequencing on the Illumina NextSeq 500 were performed by the Scripps Florida Genomics Core. De-multiplexed and quality filtered raw reads (fastq) were trimmed (adaptor sequences) using Flexbar 2.4 and aligned to the reference genome using TopHat version 2.0.9 (*Trapnell et al., 2009*). HT

seqcount version 0.6.1 was used to generate gene counts and differential gene expression analysis was performed using Deseq2 (*Anders and Huber, 2010*). DeSeq2 identified differentially expressed genes (DEGs) with a cutoff of 1.5-fold change and an adjusted p-value of less than 0.05. Paired end reads mapped to the first 30 bases of Exon21 was used to determine the ratio of Exon21a (results in SynGAP-α2) vs Exon21b (results in SynGAP-α1) splicing events.

## Cell culture

### Cell lines

HeLa Cells (Kind gift of Michael Farzan) and HEK293T Cells (Kind gift of Joseph Kissil) were cultured in DMEM media containing 10% fetal bovine serum and penicillin/streptomycin. Cell lines were originally obtained from ATCC (Manassas, VA) and were mycoplasma free.

### Primary forebrain cultures

Dissociated forebrain cultures were prepared from newborn WT and homozygous littermates of the PBM line as previously described (*Beaudoin et al., 2012*). Briefly, forebrains were isolated and incubated with a digestion solution containing papain for 25 min at 37 °C. Tissues were washed and triturated in Neurobasal medium containing 5% FBS. Cells were plated on poly-D-lysine at a density of 1000 cells per mm$^2$. Cultures were maintained in Neurobasal A media (Invitrogen) supplemented with B-27 (Invitrogen) and Glutamax (Invitrogen). At DIV4 cells were treated with FuDR to prevent glial expansion. The cells were sparsely labeled by administration of AAVs (CamKII.Cre, 10$^4$vg/ml, Addgene # 105558-AAV9 and CAG.Flex.EGFP, 10$^8$vg/ml, Addgene #28304-PHPeB) at DIV 9–10 and processed for experiments 10–11 days later.

## In situ colocalization assay

HeLa cells were plated on glass coverslips and transfected with PSD95-tRFP (Plasmid #52671, Addgene) and/or EGFP-tagged SynGAP C-terminal constructs EGFP-CCα1 or EGFP-CCPBM plasmids (made in house) were co-transfected into HeLa cells using lipofectamine 2000 according to manufacturer instructions. Cells were then fixed with 4% PFA and washed multiple times with PBS prior to mounting with Prolong Gold with DAPI (P36931, Thermo). Confocal stacks spanning entire cells were obtained using UPlanSApo 100 × 1.4 NA oil-immersion objective mounted on Olympus FV1000 laser-scanning confocal microscope using Nyquist criteria for digital imaging. Maximum intensity projections were used for the analysis. Nuclei of cells were defined by DAPI staining, and the EGFP-CC nuclear localization was calculated as the EGFP (colocalized with nucleus) / EGFP (within entire cell perimeter).

## PSD95-SynGAP co-IP assay

PSD95-tRFP (Plasmid #52671, Addgene) and/or full length EGFP-SynGAPα1/PBM (made in house) plasmids were transfected in HEK293T cells using Lipofectamine 2000. Cells were homogenized with Pierce IP Lysis buffer (87787, Thermo) containing protease & phosphatase inhibitors. Lysates were then incubated for 2 hr at RT with 1.5 mg Dynabeads (10,004D, Thermo) functionalized with 10 µg of anti-PSD95 (Thermo, MA1-045) or IgG control (ab18415, Abcam). After extensive washing, immunoprecipitated proteins were eluted with Leammeli buffer at 70 °C for 10 min with agitation. Eluted proteins were detected via western blot using PSD-95 (Thermo, MA1-045) and SynGAP (D20C7, CST) antibodies.10% of the input and 20% of IP elute were used for each sample.

## In vitro treatments

To silence neuronal activity and block NMDAR signaling, cultures were treated for 3 hr with 1 µM TTX and 200 µM APV. To induce chemical LTP, Cells were thoroughly washed and perfused with basal ECS (143 mM NaCl, 5 mM KCl, 10 mM HEPES (pH 7.42), 10 mM Glucose, 2 mM CaCl$_2$, 1 mM MgCl$_2$, 0.5 µM TTX, 1 µM Strychnine, and 20 µM Bicuculline) for 10 min. Then magnesium-free ECS containing 200 µM Glycine (or 10 µM Glycine for weak cLTP) was applied for 10 min. Cells were then washed with and incubated in basal ECS for additional 10 min prior to downstream application.

## Subcellular fractionation

### From tissue

Frozen hippocampi or cortex were homogenized using a Teflon-glass homogenizer in ice-cold isotonic solution (320 mM sucrose, 50 mM Tris pH 7.4, phosphatase & protease inhibitors). The homogenate

was then centrifuged at 1000 g for 10 min at 4 °C. The supernatant (S1) was centrifuged at 21,000 g for 30 min. The pellet (P2) was resuspended in isotonic buffer and layered on top of a discontinuous sucrose density gradient (0.8 M, 1.0 M or 1.2 M sucrose in 50 mM Tris pH 7.4, + inhibitors) and centrifuged at 82,500 g for 2 hr at 4 °C. The interface of 1.0 M and 1.2 M sucrose was collected as a synaptosomal fraction. Synaptosomes were diluted using 50 mM Tris pH7.4 ( + inhibitors) to bring the sucrose concentration to 320 mM. The diluted synaptosomes were then pelleted by centrifugation at 21,000 g for 30 min at 4 °C. The synaptosome pellet was then resuspended in 50 mM Tris pH 7.4 and then mixed with an equal part 2% Triton-X ( + inhibitors). This mixture was incubated at 4 °C with rotation for 10 min followed by centrifugation at 21,000xg for 20 min to obtain a supernatant (Syn/Tx) and a pellet (PSD).

### From primary culture

Cultured neurons (DIV 18–21), were homogenized by passage through 22 G needle 10 times in ice-cold isotonic buffer (320 mM sucrose, 50 mM Tris, protease & phosphatase inhibitor mix). Homogenates were centrifuged at $1000 \times g$ for 10 min at 4 °C. The supernatant (S1) was centrifuged at $15,000 \times g$ for 20 min at 4 °C to obtain the crude membrane (P2 fraction). The P2 pellet was resuspended with ice-cold hypotonic buffer (50 mM Tris, protease and phosphatase inhibitor mix) and was incubated for 30 min at 4 °C. Then the sample was centrifuged 21,000 x g for 30 min to obtain synaptic plasma membrane (SPM) fraction. SPM is reconstituted in hypotonic buffer then equal volume of hypotonic buffer with 2% Triton-X was added and the mixture was incubated 15 min on ice. Lysates were centrifuged at 21,000 g for 30 min at 4 °C to obtain a soluble fraction (Syn/Tx) and a pellet (PSD), which was resuspended in 50 mM Tris containing 0.5% SDS. To completely solubilize PSD fraction, we have briefly sonicated and heated samples to 95 °C for 5 min.

## Immunoblotting

Protein lysates were extracted from the hippocampi or cortices of adult mice and dissected in ice-cold PBS containing Phosphatase Inhibitor Cocktails 2 and 3 (Sigma-Aldrich, St. Louis, MO) and Mini-Complete Protease Inhibitor Cocktail (Roche Diagnostics) and immediately homogenized in RIPA buffer (Cell Signaling Technology, Danvers, MA), and stored at −80 °C. Sample protein concentrations were measured (Pierce BCA Protein Assay Kit, Thermo Scientific, Rockford, IL), and volumes were adjusted to normalize microgram per microliter protein content. For phospho-protein analysis, in vitro cultures were directly lysed with laemmeli sample buffer, sonicated and centrifuged to minimize DNA contamination. 10 µg of protein per sample were loaded and separated by SDS-PAGE on 4–15% gradient stain-free tris-glycine gels (Mini Protean TGX, BioRad, Hercules, CA), transferred to low fluorescence PVDF membranes (45 µm) with the Trans-Blot Turbo System (BioRad). Membranes were blocked with 5% powdered milk (BSA for phospho-proteins) in TBST and probed overnight at 4 °C with the following primary antibodies: Pan-SynGAP (Thermo, PA1-046), SynGAP-α1 (Millipore, 06–900), SynGAP-α2 (abcam, ab77235), SynGAP-β(Kind gift of Rick Huganir), PSD-95 (Thermo, MA1-045), Synaptophysin (Novus, NB300-653), pERK (CST, 9106), ERK (CST, 4696), GluA1 (Millipore, MAB2263), TARP (Millipore, Ab9876), LRRTM2 (Thermo Pierce, PA521097).

## Immunocytochemistry

*For SynGAP – PSD95 colocalization,* neurons were fixed in 4% PFA, 4% sucrose for 5 min at RT and treated with MetOH for 15 min at –20 °C. The cells were then washed with PBS and permeabilized in PBS 0.2% TritonX-100 for 10 min. Samples were then blocked for 1 hr and probed for SynGAP (D20C7, CST) and PSD95 (MA1-045, Abcam) overnight. After PBS washes, samples were probed with appropriate secondary antibodies for 1 hr in the dark at room temperature. The coverslips were then washed, mounted (Prolong Glass) and cured. Confocal stacks were obtained. For analysis, maximum intensity Z projection was obtained from each confocal image. Individual synapses were traced as PSD95 positive puncta selected using an arbitrary threshold which was kept constant across all images. Mean SynGAP and PSD95 signals were measured from individual synapses. *For surface GluA1 staining,* neurons were immediately fixed in ice-cold pH 7.2 4% PFA, 4% sucrose for 20 min on ice. Then, samples were washed three times with ice-cold PBS and blocked for 1 hr min in PBS containing 10% NGS. Cells were then incubated overnight with a primary antibody targeting the extracellular N terminus of GluA1 (MAB2263, Millipore) and then washed with 10% goat serum twice to remove

excess primary antibody. After PBS washes, Alexa dye–conjugated secondary antibodies were added for 1 hr in the dark at room temperature. The coverslips were then washed, mounted (Prolong Glass) and cured. Surface GluA1 levels were measured from manually traced individual dendritic spines from maximum intensity Z projection images using EGFP channel (cell fill). All confocal stacks were obtained for 6–12 individual fields from multiple coverslips per culture with UPlanSApo 100 × 1.4 NA oil-immersion objective mounted on Olympus FV1000 laser-scanning confocal microscope using Nyquist criteria for digital imaging. Forty to 80 μm stretches of secondary dendrites in neurons with pyramidal morphology were imaged.

## PSD95 immunoprecipitation and mass spectrometry

Harvested neurons were lysed in DOC lysis buffer (50 mM Tris (pH 9), 30 mM NaF, 5 mM sodium ortho-vanadate, 20 mM β-glycerol phosphate, 20 μM $ZnCl_2$, Roche complete, and 1% sodium deoxycholate). The lysate was then centrifuged at 35,000 RPM for 30 min at 4 °C and lysate containing 1 mg of protein was incubated with 2 μg Psd95 antibody (Neuromab, catalog # 75–048) at 4 °C overnight with rotation. The following day, IPs were incubated with Dynabeads protein G (Thermo Fisher Scientific, catalog # 10,004D) for 2 hours at 4 °C. IPs were then washed three times with IP wash buffer (25 mM Tris (pH 7.4), 150 mM NaCl, 1 mM EDTA, and 1% Triton X-100). Dynabeads were re-suspended in 2 X LDS sample buffer and incubated at 95 °C for 15 min for elution. The eluate was incubated with DTT at a final concentration of 1 mM at 56 °C for 1 hr followed by a 45-min room temperature incubation with Iodoacetamide at a final concentration of 20 mM.

Samples were loaded onto 4–12% Bis-Tris gels and separated at 135 V for 1.5 hr. Gels were stained with InstantBlue (Expedeon, catalog # 1SB1L) to visualize bands. The heavy and light chains of Immunoglobulin were manually removed. Gels were then destained using 25% ethanol overnight. Gel lanes were cut, individual gel slices were placed into 96 well plates for destaining, and peptide digestion was completed at 37 °C overnight. Peptides were extracted with acetonitrile, dried down, and then desalted using stage tips. All LC-MS experiments were performed on a nanoscale UHPLC system (EASY-nLC1200, Thermo Scientific) connected to an Q Exactive Plus hybrid quadrupole-Orbitrap mass spectrometer equipped with a nanoelectrospray source (Thermo Scientific). Samples were resuspended in 10 μL of Buffer A (0.1% FA) and 2 μL were injected. Peptides were separated by a reversed-phase analytical column (PepMap RSLC C18, 2 μm, 100 Å, 75 μm X 25 cm) (Thermo Scientific). Flow rate was set to 300 nl/min at a gradient starting with 3% buffer B (0.1% FA, 80% acetonitrile) to 38% B in 110 min, then ramped to 75% B in 1 min, then ramped to 85% B over 10 min and held at 85%B for 9 min. Peptides separated by the column were ionized at 2.0 kV in the positive ion mode. MS1 survey scans for DDA were acquired at resolution of 70 k from 350 to 1800 m/z, with maximum injection time of 100ms and AGC target of 1e6. MS/MS fragmentation of the 10 most abundant ions were analyzed at a resolution of 17.5 k, AGC target 5e4, maximum injection time 65ms, and an NCE of 26. Dynamic exclusion was set to 30 s and ions with charge 1 and >6 were excluded. The maximum pressure was set to 1180 bar and column temperature was constant at 50 °C. Proteome Discoverer 2.2 (Thermo Fisher Scientific) was used to process MS data and analyzed using Sequest HT against Uniprot mouse databases combined with its decoy database. With respect to analysis settings, the mass tolerance was set 10 parts per million for precursor ions and 0.02 daltons for fragment ions, no more than two missed cleavage sites were allowed, static modification was set as cysteine carbamidomethylation, and oxidation of methionine was set as a dynamic modification. False discovery rates (FDRs) were automatically calculated by the Percolator node of Proteome Discoverer with a peptide and protein FDR cutoff of 0.01. Label-free quantification was performed using Minora node in Proteome Discoverer. Abundances of identified PSD95 interacting proteins in WT and mutant neurons were compared using relative abundances such that proteins with a fold change in abundance ratio of >2.0 or < 0.5 were considered to be differentially associated to PSD95.

## Hippocampal LTP and extracellular recordings

Acute transverse hippocampal slices (350 μm) were prepared using a Leica Vibroslicer (VT 1000 S), as described previously (*Babayan et al., 2012*). Slices were cut into ice cold, high magnesium artificial cerebrospinal fluid (aCSF) solution containing in mM: 124 NaCl, 3 KCl, 1.25 $KH_2PO_4$, 5 $MgSO_4$, 26 $NaHCO_3$, and 10 dextrose. Slices were then transferred to an interface recording chamber maintained at 31°C ± 1°C, oxygenated in 95% $O_2$ / 5% $CO_2$ and constantly perfused (60–80 ml/hr) with recording

aCSF (in mM: 124 NaCl, 3 KCl, 1.25 $KH_2PO_4$, 1.5 MgSO4, 2.5 $CaCl_2$, 26 $NaHCO_3$, and 10 dextrose). Slices equilibrated in the chamber for approximately 2 hr before experimental use. Field excitatory postsynaptic potentials (fEPSPs) were recorded from CA1b stratum radiatum using a single glass pipette filled with 2 M NaCl (2–3 MΩ). Bipolar nicrome stimulating electrodes (65 µm diameter, A-M Systems) were positioned at two sites (CA1a and CA1c) in the apical Schaffer collateral-commissural projections to provide activation of separate converging pathways of CA1b pyramidal cells. Pulses were administered in an alternating fashion to the two electrodes at 0.05 Hz using a current that elicited a 50% maximal population spike-free response. After establishing a 10–20 min stable baseline, long-term potentiation (LTP) was induced in the experimental pathway by delivering 7 'theta' bursts, with each burst consisting of four pulses at 100 Hz and the bursts themselves separated by 200 ms (i.e. theta burst stimulation or TBS). The stimulation intensity was not increased during TBS. The control pathway received baseline stimulation (0.05 Hz) to monitor the health of the slice. The fEPSP slope was measured at 10%–90% fall of the slope and all values pre- and post- TBS were normalized to mean values for the last 10 min of baseline recording. Baseline measures for all groups included paired-pulse facilitation and input/output curves. Recordings were digitized at 20 kHz using an AC amplifier (A-M Systems, Model 1700) and collected using NAC 2.0 Neurodata Acquisition System (Theta Burst Corporation).

## Ex vivo whole-cell electrophysiology

Acute coronal slices (350 µm) were prepared from 10 to 14 days old mice for 3 mouse lines. Ice-cold cutting solution was used for slice preparation and contained the following (in mM): 119 NaCl, 2.5 KCl, 1.3 MgSO4, 2.5 CaCl2, 1 NaH2PO4, 11 D-glucose and 26.3 NaHCO3, pH 7.4, 300–310 mOsm bubbled with 95%CO2 and 5%O2. The slices were then warmed to 37 °C for an hour approximately in standard artificial cerebrospinal fluid (aCSF), composed of (mM): 125 NaCl, 2.5 KCl, 24 NaHCO3, 2 CaCl2, 1.25 NaH2PO4, 2 MgSO4, and 10 D-Glucose, and equilibrated with 95% O2 and 5% CO2 (pH 7.4, ~ 300 mOsm). Following this, slices were maintained in bubbled aCSF at room temperature until transferred to a submerged-type recording chamber (Warner Instruments, Hamden, CT). All experiments were performed at 32 °C ± 2 (perfusion rate of 2–3 mL/min). Whole-cell patch clamp experiments were conducted from visually identified L2/3 neurons using infrared DIC optics. L2/3 excitatory cells were identified by their soma shape and their location ~150 uM below the L1-L2 boundary. Regular spiking was confirmed in current clamp and miniature excitatory postsynaptic current (mEPSC) were recorded from identified cells for 5 sweeps each lasting a minute, using the following internal solution (in mM): 120 CsCl, 10 K-HEPES, 10 EGTA, 5 QX314-Br, 4 Mg-ATP, 0.3 Na-GTP, 4 MgCl2 (pH 7.3, 290–295 mOsm). Perfusion solution aCSF was supplemented with 100 µM picrotoxin and 1 µM TTX. Cells with access resistance >20 MΩ or were unstable ( > 20% change) were discarded from further analysis. Recordings were made using borosilicate glass pipettes (3–6 MΩ; 0.6 mm inner diameter; 1.2 mm outer diameter; Harvard Apparatus). All signals were amplified using Multiclamp 700B (Molecular Devices, Sunnyvale, CA), filtered at 4 KHz, digitized (10 KHz), and stored on a personal computer for off-line analysis. Analog to digital conversion was performed using the Digidata 1,440 A system (Molecular Devices). Data acquisition and analyses were performed using pClamp 11.2software package (Clampex and Clampfit programs; Molecular Devices) and minianalysis (Synaptosoft). The events were considered mini-EPSCs if the peak of an event was >5 pA.

## Behavior

At weaning, four mice were randomly allocated to one cage with respect to genotype with males and females being housed separately. Randomization of cage allocation was restricted in that, as much as possible, mice from the same litter were placed in different cages so that no single litter was over-represented in any single experiment. Cages utilized for behaviors contained cardboard pyramidal-shaped huts with two square openings on opposing sides of the hut for the purposes of environmental enrichment and to assist with transfers from home cages to behavioral apparatuses. All mice were handled for several minutes on three consecutive days prior to commencement of behavioral testing. Tails were marked for easy identification and access from home cages during testing. Experimenters were blind to mouse genotype while conducting all tests.

## Flurothyl-induced seizures

Flurothyl-induced seizure studies were performed based on prior studies with some modifications (*Ozkan et al., 2014*; *Clement et al., 2012*; *Dravid et al., 2007*). Briefly, experiments were conducted in a chemical fume hood. Mice were brought to the experimental area at least 1 hr before testing. To elicit seizures, individual mice were placed in a closed 2.4 L Plexiglas chamber and exposed to 99% Bis (2,2,2-triflurothyl) ether (Catalog# 287571, Sigma-Aldrich, St. Louis, MO). The flurothyl compound was infused onto a filter paper pad, suspended at the top of the Plexiglas chamber through a 16 G hypodermic needle and tube connected to a 1 ml BD glass syringe fixed to an infusion pump (KD Scientific, Holliston, MA, USA, Model: 780101) at a rate of 0.25 ml/min. The infusion was terminated after the onset of a hind limb extension that usually resulted in death. Cervical dislocation was performed subsequently to ensure death of the animal. Seizure threshold was measured as latency (s) from the beginning of the flurothyl infusion to the beginning of the first myoclonic jerk.

## Morris water maze

Mice were run in a standard comprehensive Morris water maze paradigm including a cue test with a visual platform and an acquisition protocol with a hidden platform. All phases of the paradigm were run in a dedicated water maze room in the Scripps Florida Mouse Behavior Core. A water maze system including a plastic white opaque pool (Cat# ENV-594M-W, Med Associates), measuring ~122 cm diameter at the water surface, supported by a stand (ENV-593M-C) and equipped with a floor insert (ENV-595M-FL) covering a submerged heater was utilized for all water maze experimentation. An adjustable textured platform (17.8 cm diameter, ENV-596M) was placed atop the floor insert in one of two different quadrants, depending on the specific phase of the paradigm (NW quadrant for initial training and probe test and SE quadrant for reversal training and probe tests), for mice to escape the water. Water temperatures were controlled to 22.5°C ± 0.5 °C using a built-in heater and monitored with a digital temperature probe. This water temperature motivated the mice to escape the water without eliciting hypothermic conditions. The tank was emptied, cleaned and refilled once every three days to avoid unsafe accumulation of bacteria. Water was made opaque by the addition of a white opaque non-toxic paint (Crayola) forcing mice to utilize extra-maze cues when locating the hidden platform (0.5 cm beneath the surface of the water). These spatial cues (large black cardboard circle, star, square, white X on black background) were placed on the walls of the room at different distances from the pool. The pool edge was demarcated with directional units (W, N, E, S) to aid assignment of invisible platform 'quadrants' to the pool arena outlined by the video tracking system. Various strip lights were positioned on the walls near the ceiling to allow for a moderate level of lighting (200 lux), enough for the mice to see the extra-maze cues adequately without eliciting undue anxiety. Thirty minutes prior to commencement of daily trials, the lights and heater were turned on, and mouse home cages were placed on heating pads on a rack in the water maze room to provide a warm place for the mice between trials. Cage nestlets were replaced with strips of paper towels to better facilitate drying after trials. Mice were monitored during trials for signs of distress and swimming competence. None of the mice tested had swimming issues, and floating was discouraged with gentle nudges. Mice received four trials per day during cue and acquisition phases and one trial per day for probe trials. Three cages (12 mice) were run at a time such that ITIs for each day lasted about 20 min with trial duration lasting until the mouse found the platform or a maximum of 60 s. Each trial commenced when the mouse was automatically detected in the pool by the tracking system (Ethovision, Noldus). Each mouse was lowered into the pool facing its edge at one of the four directional units (W, N, E, S) in a clockwise manner, with the first of the four trials starting closest to the platform ('NW quadrant'), which was positioned in the central area of the quadrant dictated by the tracking system. This same series of daily trial commencements were followed for all mice for each of the cue tests, acquisition protocol, and reversal protocol. If the mouse did not locate the platform in 60 s, the experimenter's hand guided them to the platform. Because the mice are eager to escape the water, the mice quickly learned to follow hand direction to the platform, minimizing physical manipulation of the animals during the trials. Mice were allowed 15 s on the platform at the end of each trial before being picked up, dried with absorbent wipes, and placed back into their warmed home cage.

On the first day of testing, mice were given a cue test with the platform positioned just above the surface of the water and a metal blue flag placed upon it for easy visual location of the platform. This test allows for detection of individual visual and swimming-related motor deficits and allows the mice

to habituate to the task (climbing on the platform to escape the water). The platform was placed in a different location for each of the four trials with spatial cues removed by encirclement of the pool with a white plastic curtain.

On the next day, acquisition trials began with the hidden platform remaining in the same location ("NW quadrant") for all trials/days and the curtain drawn back for visibility of the spatial cues. Several measures (distances to platform) and criteria to reach the platform (approximately 90% success rate, approximately 20 s latency to find platform) during the acquisition phases were recorded and achieved before mice were deemed to have learned the task. The performances of the four trials were averaged for each animal per day until criteria were met.

### Open-field test

Naive mice were individually introduced into one of eight adjacent open-field arenas for 30 min and allowed to explore. Open field arenas consisted of custom made clear acrylic boxes (43 × 43 × 32 h cm) with opaque white acrylic siding surrounding each box 45 × 45 × 21.5 h cm to prevent distractions from activities in adjacent boxes. Activity was monitored with two CCTV cameras (Panasonic WV-BP334) feeding into a computer equipped with Ethovision XT 11.5 for data acquisition and analyses. A white noise generator (2325–0144, San Diego Instruments) was set at 65 dB to mask external noises and provide a constant noise level. Fluorescent linear strip lights placed on each of the four walls of the behavioral room adjacent to the ceiling provided a lower lighting (200 lux) environment than ceiling lighting to encourage exploration.

### Contextual fear conditioning

A dedicated fear conditioning room in the TSRI Florida Mouse Behavior Core contains four fear conditioning devices that can be used in parallel. Each apparatus was an acrylic chamber measuring approximately 30 × 30 cm (modified Phenotyper chambers, Noldus, Leesburg, VA). The top of the chamber is covered with a unit that includes a camera and infrared lighting arrays (Noldus, Ethovision XT 11.5, Leesburg, VA) for monitoring of the mice. The bottom of the chamber is a grid floor that receives an electric shock from a shock scrambler that is calibrated to 0.40 mA prior to experiments. The front of the chamber has a sliding door that allows for easy access to the mouse. The chamber is enclosed in a sound-attenuating cubicle (Med Associates) equipped with a small fan for ventilation. Black circular, rectangular and white/black diagonal patterned cues were placed outside each chamber on the inside walls of the cubicles for contextual enhancement. A strip light attached to the ceilings of the cubicles provided illumination. A white noise generator (~65 dB) was turned on and faced toward the corner of the room between the cubicles. The fear conditioning paradigm consisted of two phases, training, followed by testing 1 and 26, or 30 d thereafter. The 4.5 min training phase consisted of 2.5 min of uninterrupted exploration. Two shocks (0.40 mA, 2 s) were delivered, one at 2 min 28 s, the other at 3 min and 28 s from the beginning of the trial. During testing, mice were placed into their designated chambers and allowed to roam freely for 5 min. Immobility durations (s) and activity (distances moved (cm)) during training and testing were obtained automatically from videos generated by Ethovision software. Activity suppression ratio levels were calculated: 0–2 min activity during testing/0–2 min activity during training +testing.

## Acknowledgements

This work was supported in part by NIH grants from the National Institute of Mental Health (MH096847 and MH108408 to G.R., MH115005 and MH113949 to M.P.C, and MH105400 to C.A.M.), the National Institute for Neurological Disorders and Stroke (NS064079 to G.R.), the Eunice Kennedy Shriver National Institute of Child Health and Human Development (HD089491 to G.L.), the National Institute for Drug Abuse (DA034116 and DA036376 to C.A.M.), the Spanish Ministerio de Ciencia, Innovación y Universidades RTI2018-097037-B-I00 MINECO/MCI/AEI/FEDER, EU, Award AC17/00005 by ISCIII through AES2017 and within the NEURON framework, Ramón y Cajal Fellowship (RYC-2011–08391 p), IEDI-2017–00822 and AGAUR SGR14-297 and 2017 SGR 1776 (to A.B.). M.K. was supported by Autism Speaks Weatherstone Pre-Doctoral fellowship (10646). G.G. was supported by a predoctoral fellowship from the Spanish Ministerio de Educación (BES-2013–063720). V.A. was supported by a training fellowship from the Leon and Friends Charitable Foundation.

## Additional information

### Funding

| Funder | Grant reference number | Author |
|---|---|---|
| National Institute of Mental Health | MH096847 | Gavin Rumbaugh |
| Autism Speaks | 10646 | Murat Kilinc |
| National Institute of Mental Health | MH108408 | Gavin Rumbaugh |
| National Institute of Child Health and Human Development | HD089491 | Gary Lynch |
| Eunice Kennedy Shriver National Institute of Child Health and Human Development | HD089491 | Gary Lynch |
| Ministerio de Ciencia, Innovación y Universidades | IEDI-2017-00822 | Àlex Bayés |
| National Institute for Neurological Disorders and Stroke | NS064079 | Gavin Rumbaugh |
| National Institute for Drug Abuse | DA034116 | Courtney A Miller |
| National Institute for Drug Abuse | DA036376 | Courtney A Miller |
| Ministerio de Ciencia, Innovación y Universidades | AGAUR SGR14-297 | Àlex Bayés |
| Ministerio de Ciencia, Innovación y Universidades | 2017 SGR 1776 | Àlex Bayés |
| Spanish Ministerio de Educación | BES-2013-063720 | Gemma Gou |
| Leon and Friends Charitable Foundation | | Vineet Arora |
| National Institute of Mental Health | MH108408 | Gavin Rumbaugh |
| National Institute of Mental Health | MH115005 | Marcelo Coba |
| National Institute of Mental Health | MH113949 | Marcelo Coba |
| National Institute of Mental Health | MH105400 | Courtney A Miller |

The funders had no role in study design, data collection and interpretation, or the decision to submit the work for publication.

### Author contributions

Murat Kilinc, Conceptualization, Data curation, Investigation, Methodology, Validation, Visualization, Writing - original draft, Writing - review and editing; Vineet Arora, Formal analysis, Investigation, Methodology, Visualization; Thomas K Creson, Investigation, Methodology, Supervision; Camilo Rojas, Data curation, Formal analysis, Investigation, Methodology, Visualization; Aliza A Le, Formal analysis, Investigation, Methodology; Julie Lauterborn, Formal analysis, Investigation, Methodology, Writing - review and editing; Brent Wilkinson, Conceptualization, Data curation, Formal analysis; Nicolas Hartel, Data curation, Formal analysis, Investigation; Nicholas Graham, Data curation, Investigation, Visualization; Adrian Reich, Methodology, Visualization; Gemma Gou, Formal analysis; Yoichi Araki, Resources, Validation; Àlex Bayés, Formal analysis, Investigation, Project administration, Writing - review and

editing; Marcelo Coba, Data curation, Formal analysis, Investigation, Methodology, Project adminis-
tration, Visualization, Writing - review and editing; Gary Lynch, Formal analysis, Investigation, Meth-
odology, Project administration, Writing - review and editing; Courtney A Miller, Writing - review
and editing; Gavin Rumbaugh, Conceptualization, Formal analysis, Funding acquisition, Investigation,
Project administration, Supervision, Writing - original draft

### Author ORCIDs
Murat Kilinc http://orcid.org/0000-0003-0928-7573
Vineet Arora http://orcid.org/0000-0001-7856-0401
Yoichi Araki http://orcid.org/0000-0002-3455-9377
Àlex Bayés http://orcid.org/0000-0002-5265-6306
Gavin Rumbaugh http://orcid.org/0000-0001-6360-3894

### Ethics
This study was performed in strict accordance with the recommendations in the Guide for the Care
and Use of Laboratory Animals of the National Institutes of Health. All of the animals were handled
according to approved institutional animal care and use committee (IACUC) protocols (#15-037 and
#15-038) of Scripps Florida.

### Decision letter and Author response
Decision letter https://doi.org/10.7554/eLife.75707.sa1
Author response https://doi.org/10.7554/eLife.75707.sa2

## Additional files

### Supplementary files
• Supplementary file 1. Summary of Phenotypes in Syngap1 Isoform Mice: Summary of
electrophysiological, behavioral, and protein expression data for the three new Syngap1 mouse lines
presented in this study.

• Transparent reporting form

### Data availability
All data generated or analysed during this study are included in the manuscript and supporting file;
Source Data files have been provided for western blots and mass spec experiments.

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
