## [Editor Report]

This study used three different mouse lines with altered expression of splice variants of SynGAP1 and reports that SynGAP1-α variants are more important than the SynGAP1-β variants for the regulation of cognitive function and seizure protection in mice. Given the well-known importance of the SYNGAP1 mutations in the pathophysiology of neurodevelopmental disorders, and the key regulatory roles of SynGAP1 for excitatory synaptic functions, these results provide timely and comprehensive data supporting the in vivo functions of individual SynGAP1 splice variants, including the α-1/2 variants, and suggests the therapeutic potential of increasing specific SynGAP1-α variants.

---

## [Decision Letter]

**Decision letter after peer review:**

[Editors’ note: the authors submitted for reconsideration following the decision after peer review. What follows is the decision letter after the first round of review.]

Thank you for submitting your work entitled "SynGAP C-terminal splicing enables isoform-specific tuning of NMDA receptor signaling linked to cognitive function" for consideration by *eLife*. Your article has been reviewed by 3 peer reviewers, one of whom is a member of our Board of Reviewing Editors, and the evaluation has been overseen by Gary Westbrook as the Senior Editor. The following individual involved in review of your submission has agreed to reveal their identity: Shigeo Okabe (Reviewer #3).

Our decision has been reached after consultation between the reviewers. Based on these discussions and the individual reviews below, we regret to inform you that your work will not be considered further for publication in *eLife*.

Although the reviewers find your manuscript novel and comprehensive, they think that additional experiments are required to fully support the main conclusions. We realize that the current corona situation makes it very difficult to perform additional experiments in the near future. However, if you decide to address the review comments with additional experiments when able, we are willing to consider a resubmission of the manuscript.

*Reviewer #1:*

This study reports the role of the PBM (PDZ-binding motif) of SynGAP in the regulation of synaptic structure, signaling, and transmission. Moreover, this study shows that the SynGAP PBM is important for setting the threshold for the dispersal of SynGAP induced by NMDAR activation. In addition, the deletion of PBM suppresses STP and LTP and impairs spatial learning, contextual fear memory, spontaneous alteration without affecting anxiety-related behavior.

SynGAP is known to interact with PSD-95 in the postsynaptic density (PSD), but the function of this interaction has remained unclear. This study addresses this question using a mouse genetic approach and various in vitro and in vivo experiments. In addition, the results are of high quality and largely convincing. I think this is an important contribution to the understanding of SynGAP functions as well as the roles of the PBMs and their interactions with synaptic PDZ proteins.

1. The deletion of the SynGAP PBM leads to suppressed STP and LTP. I am wondering whether LTD and mGluR-LTD are also impacted because LTP and LTD are known to interfere with each other, and altered synaptic signaling might also affect LTD and mGluR-LTD, which may also affect neuronal functions.

2. The roles of the SynGAP PBM on unitary functions of excitatory synapses remain largely unclear, which can easily be addressed by measuring mEPSCs at mutant hippocampal synapses. The increased spine diameter and surface GluA1 levels in Figure 2A suggests this possibility.

*Reviewer #2:*

In this study, Kilinc et al. investigate the function of the SynGAP1 PDZ-binding motif, present in the α1 isoform, by characterizing a novel mouse model with PBM inactivating mutations. As expected, the PBM mutations in SynGAP1 abolish its interaction with PSD-95, whilst maintaining wild-type levels of the protein. Neurons from Syngap1PBM/PBM mice have reduced SynGAP1 content in dendritic spines and in PSD fractions and exhibit a range of synaptic and excitability phenotypes strikingly reminiscent of neurons from Syngap1 knockout mice. Interestingly, many of the synaptic phenotypes are rescued by NMDA receptor blockade, suggesting that NMDAR-dependent signalling causes part of the dysfunction in these neurons. In addition to altering PSD composition, neurons with the mutant SynGAP1 isoforms have altered LTP and are proposed to have a reduced threshold for NMDAR-dependent signalling. Together these phenotypes cause deficits in behaviours affecting learning and memory, similar to full Syngap1 knockouts, revealing the importance of the PBM in cognitive function.

This manuscript is very well written and logically presented, addressing fundamental questions about SYNGAP1, a critically important gene in human neurodevelopment disorders and synaptic structure and function. However, despite the clear synaptic and behavioural phenotypes in the mouse model, it is still somewhat unclear how dysfunction of the α1 isoform causes these disturbances. If further experiments can solidify the proposed this mechanism, I would highly recommend this work for publication in *eLife*.

1) One of the central posits in this manuscript is that mutation of the PBM causes increased Syngap1 mobility and lowers the threshold for NMDA receptor-dependent signalling. However, the evidence for this appears to be insufficient, relying mostly on a single experiment (Figure 3D-J). Further work to bolster these results would be highly valuable, given that discovering the function of the PBM is one of central the aims of this paper.

- The increased SynGAP1 mobility in PBM deficient neurons and after wcLTP is suggested by a reduced abundance in the PSD fraction, but this is an indirect measure. Other experiments should be designed to conclusively show this. For instance, fluorescence recovery after photobleaching (FRAP) can be used to directly measure the mobility of GFP-tagged proteins in dendritic spines. Alternatively, the wcLTP protocol could be used in cultured neurons to measure the dispersion of syngap1 out of dendritic spines or PSD95 puncta by immunostaining, similar to Figure 2 Supplement 1A-B. This experiment may even be improved by using α1-specific antibody for immunostaining if this is possible.

- The experiments showing that PBM mutations lowers the threshold for NMDA receptor-dependent signalling should also be strengthened. Can these experiments be replicated with a different NMDA-dependent stimulus? Does 10uM glycine have a functional effect on PBM mutant neurons? I.e. Is there a change in spine size, surface GluA1 levels or electrophysiological properties?

2) It is difficult to appreciate how important the α1 isoform is without knowing how abundant it is compared to other isoforms in the brain. Can the authors show the relative expression of each isoform at the mRNA and/or protein level?

3) The mobility of the SynGAP1 protein appears to be a critical to its function. However, a comprehensive discussion on why SynGAP1 localization is altered after NMDAR activation, and how defects in this process leads to the observed synaptic phenotypes is lacking in the discussion. In addition, how does the movement of SynGAP1 out of the PSD affect downstream GTPase signalling? What is the proposed mechanism for how the blockade of NMDA receptors rescues the localization and synaptic phenotypes in neurons from Syngap1PBM/PBM mice? All of these themes are brought up in the paper but are not sufficiently discussed and detailed.

*Reviewer #3:*

This manuscript by Kilinc et al. reports the plasticity-related function of SynGAP C-terminal domain in SynGAP alpha1 isoform by generating a genetically modified mouse line containing critical point mutations in the exon 21b, which disrupt the interaction between SynGAP alpha1 isoform and PSD-95. The specific alterations in the C-terminal domain induced impairment of LTP in the hippocampus and also memory-related tasks. Experiments in cultured neurons provided additional cell biological insights into the regulation of synaptic plasticity by the SynGAP alpha1 C-terminal domain. This reviewer positively evaluates the authors' findings that link the specific splice variant of SynGAP to synaptic function and memory-related circuit function. However, several points should be corrected and improved in a future manuscript.

1. The authors found increased sensitivity to weak glycine-dependent NMDAR activation in PBM/PBM neurons, which resulted in the phosphorylation of GluA1. This finding is seemingly inconsistent with the slice electrophysiology data of reduced short-term and long-term potentiation. To link these two experiments, it will be helpful to check the response of acute slices to weak stimulation (weak stimulation can induce more responses in PBM/PBM slices?) and also evaluate the level of SynGAP in PSD fraction after strong glycine-dependent NMDAR activation (less GluA1 phosphorylation in PBM/PBM neurons in culture?).

2. The underlying hypothesis of using strong and weak cLTP induction is that the difference in glycine concentration induces a different level of increase in AMPAR surface expression in culture. The validity of this hypothesis is not tested in this manuscript. Another possibility is that lower glycine concentration induces different types of plasticity, such as LTD, which is known to be easier to induce in cultured neurons. Additional information about the level of surface GluA1 or its phosphorylation between two glycine concentrations will support the hypothesis.

3. The basic synaptic transmission is enhanced in the perforant path-dentate gyrus granule cell synapses (Figure 2B). Still, the responses to the burst responses in the Schaffer-collateral-CA1 pyramidal neurons synapses were similar between genotypes (Figure 4A). It is desirable to show the input-output relationship in the Schaffer-collateral-CA1 pyramidal neurons synapses to test the presence of any alterations in the synaptic efficacy at the basal level.

4. The authors propose that the "mobility" of the SynGAP-PBM mutant is enhanced, but no direct evidence for the alterations in the mobility is presented. It is more neutral to describe the result as "a reduced affinity" to the PSD. Otherwise, additional experiments for the measurement of protein mobility will be required.

5. In Figure Supplement 1, the authors evaluated the specificity of the 06-900 antibody to the alpha1 isoform. A similar evaluation of two additional antibodies for alpha2 and β used in Figure 3 will be informative.

6. In some cases, the difference between wild type and PBM/PBM mice is moderate, with p-values more than 0.01 (Figure 2F, H, Figure 3F). It is advisable to perform a power analysis to estimate the adequate sample size required for these experiments. It is also informative to show the confidence interval instead of the p-value. (See *eLife* 2019;8:e48175 and eNeuro. 2019 Aug 1;6(4).)

[Editors’ note: further revisions were suggested prior to acceptance, as described below.]

Thank you for re-submitting your article "Endogenous Syngap1 Α Splice Forms Promote Cognitive Function and Seizure Protection" for consideration by *eLife*. Your article has been reviewed by 3 peer reviewers, and the evaluation has been overseen by Gary Westbrook as the Senior Editor. The reviewers have opted to remain anonymous.

The reviewers were supportive of your work and have discussed their reviews with one another. The Senior Editor has drafted a summary to help you prepare a revised submission. The points listed under essential revisions were extracted from the full reviews and require a response.

Essential revisions:

1. One of the key messages of this manuscript is the amount of α isoforms, but not the total amount of SynGAP, is critical in multiple behavioral phenotypes in mutant mice. In order to discuss the importance of isoforms, essential information is a relative abundance of each isoform in both wild-type and mutant mice. Throughout this manuscript, the comparisons within isoforms were performed, but no information about relative abundance between isoforms was provided. It is also unclear there are any differences in the relative abundance of isoforms in different brain regions studied in this manuscript, including the forebrain, hippocampus, and somatosensory cortex. The estimate of relative abundance should be provided, or any technical difficulty in this estimation should be explained.

2. If alpha1 is the only isoform expressed in the brain, the interpretation of introducing PSD95-nonbinding mutant is straightforward, but in the presence of other isoforms, dominant-negative effects of PBM mutant should be considered. It is possible that PBM mutant complexed with different isoforms impair localization or function of other isoforms, leading to an effect similar to the reduction of total SynGAP protein. Clarification of this point by additional experimental data is necessary.

3. Figure 9A-C. The sample size (n = 8) for mEPSC measurements in the PBM/PBM mice seems small to draw a safe conclusion, and thus should be increased, or the authors should explicitly state the limits associated with interpretation of this sample size.

4. It is unclear why the authors selected Morris water maze as the second behavioral test, which is not designed to evaluate long-lasting memory. The rationale behind selecting two distinct memory-related behavioral tests, remote contextual memory test and water maze-based training performance, should be explained.

5. The phenotype in the Morris water maze is difficult to interpret. SynGAP heterozygote mice show impairment, but IRES-TD mice showed no change. This is inconsistent with other behavioral tests. Β* mice showed improvement in the training session, but the 24 h probe test did not show a difference. It is difficult to connect the data with the result of contextual fear memory. This point should be discussed.

6. While Synap1 is expressed mainly in excitatory neurons, it would be useful if the authors included some single-cell transcriptomic data from the literature and public databases (like Allen Brain Atlas) and discussion on the relative distribution of Syngap1 in neuronal cell types in the cortex.

7. Syngap1 is a GAP for both Ras and Rap GTPases, which have opposite effects on dendrites and synapses. The authors assessed Ras activity through p-ERK/ERK Western blots. It would be useful if they also blot with P-BRaf/BRaf to assess Rap activity. Knowing their relative activity would help further interpret the data.

8. A table with all mutants and phenotypes could help the readers.*Reviewer #1:*

This study reports that different splice variants of SynGAP1 differentially regulate cognitive function and seizure protection in mice. The authors used three different mouse lines and systematically analyzed the expression patterns of different SynGAP1 splice variants in the brain and their impacts on cognitive function and seizure protection. The authors also analyzed excitatory synaptic transmission and synaptic plasticity in these mice and attempted proteomic analyses to better understand the underlying mechanisms. The overall conclusion of the study is that the SynGAP1-α variants are more important than the SynGAP1-β variants for the regulation of cognitive function and seizure protection in mice.

1. Figure 9A-C. The n numbers (n = 8) for mEPSC measurements in the PBM/PBM mice seem to be small to draw a safe conclusion, and thus should be increased.*Reviewer #2:*

Murat Kilinc et al. studied behavioral phenotypes of three Syngap1 mutant mice in order to identify the roles of distinct SYNGAP C-terminal variants. The first mutant mouse (IRES-TD mouse) with insertion of IRES-TDtomato cassette in the last exon (exon21) showed elevated open field activity, reduced time in the induction of seizure by flurothyl, less freezing behavior in remote contextual fear memory recall, and increase in frequency and amplitude of mEPSCs in L2/3 somatosensory neurons. In addition, IRES-TD mice showed a reduction in total SynGAP protein, alpha1, and alpha2 isoforms and increased β isoform in the forebrain. The second mutant mouse (β* mouse) with a mutation in the early splice acceptor of exon19 showed a slight reduction in open field activity, increase in time for seizure induction, no change in freezing behavior in remote contextual memory recall, and decrease in frequency and amplitude of mEPSCs in L2/3 somatosensory neurons. Β* mice showed no change in total SynGAP protein, a modest increase in alpha2, and a reduction in β in the forebrain. The third mutant mouse (PBM mouse) has point mutations in exon21 for disrupting the function of the PDZ-binding motif (PBM) of SynGAP to interact with PSD95. This mutant showed elevated open field activity, reduced time in the induction of seizure, less freezing behavior in remote contextual memory recall, and increased frequency and amplitude of mEPSCs in L2/3 somatosensory neurons. PBM mice showed no change in total SynGAP, alpha2, and β proteins but reduced immunoblot signal of alpha1 in the forebrain. This reduced immunoblot signal was proposed to be related to less sensitivity of the alpha1 specific antibody to the mutated isoform. By thorough behavioral and physiological analyses of three independent mutant mouse lines, the authors conclude that the level of α isoforms, rather than the total SynGAP, is critical in several neural circuit functions related to cognition and seizure protections. Previous studies revealed multiple roles of SynGAP in neural circuit development and activity-dependent regulations. This new study provides important information about the isoform-specific functions of SynGAP, especially the critical role of α isoform content for proper synapse and circuit activity.

The results obtained from multiple mutant mouse lines support the key claims of the paper. Still, there are several points that may require more experimental data or otherwise changing the interpretation of the data.

1. One of the key messages of this manuscript is the amount of α isoforms, but not the total amount of SynGAP, is critical in multiple behavioral phenotypes in mutant mice. In order to discuss the importance of isoforms, essential information is a relative abundance of each isoform in both wild-type and mutant mice. Throughout this manuscript, the comparisons within isoforms were performed, but no information about relative abundance between isoforms was provided. It is also unclear there are any differences in the relative abundance of isoforms in different brain regions studied in this manuscript, including the forebrain, hippocampus, and somatosensory cortex. The estimate of relative abundance should be provided, or any technical difficulty in this estimation should be explained.

2. The phenotypic similarity between IRES-TD mice and PBM mutant mice suggests critical roles of α isoforms in multiple behavioral phenotypes. However, the effect of PBM mutant still needs to be carefully evaluated. If alpha1 is the only isoform expressed in the brain, the interpretation of introducing PSD95-nonbinding mutant is straightforward, but in the presence of other isoforms, dominant-negative effects of PBM mutant should be considered. It is possible that PBM mutant complexed with different isoforms impair localization or function of other isoforms, leading to an effect similar to the reduction of total SynGAP protein. Clarification of this point by additional experimental data is necessary. For example, in Figure 6A, did the PSD fraction from the PBM mutant contain a similar amount of alpha2 and β isoforms? Also, in Figure 7, native PSD95 complexes from Syngap1PBM mutants were characterized. In the PSD95 complexes, was the amount of other SynGAP isoforms unchanged?

3. The authors selected remote contextual fear conditioning coupled with protein analysis in the forebrain. This selection may be based on the motivation of finding deficits in forebrain-related long-lasting memory in mutant mice. It is unclear why the authors selected Morris water maze as the second behavioral test, which is not designed to evaluate long-lasting memory. The rationale behind selecting two distinct memory-related behavioral tests, remote contextual memory test and water maze-based training performance, should be explained. In addition, the phenotype in the Morris water maze was difficult to interpret. SynGAP heterozygote mice show impairment, but IRES-TD mice showed no change. This is inconsistent with other behavioral tests. Β* mice showed improvement in the training session, but the 24 h probe test did not show a difference. It is difficult to connect the data with the result of contextual fear memory. This point should be discussed.

Line 235 Figure 4F shows a large variety in data distribution, and it is not possible to say that β mutant had "no impact" on feezing behavior.

Line 321 The rationale for NMDAR signaling suppression should be stated clearly. in vivo, there is no NMDAR signaling inhibition, and the condition without APV should be closer to the physiological situation.

Line 330 The reason why TARPs and LRRTM2 are analyzed here should be stated.

Line 386 The sentence "In this study, we created three distinct mouse lines" is not consistent with the sentence in Line 119 "We previously reported the generation of a Syngap1 mouse line with an insertion of an IRES TDtomato (IRES-TD) cassette within the 3'-UTR to facilitate endogenous reporting of active Syngap1 mRNA translation in cells [29]."

Line 388 The sentence "without appreciable change in total GynGAP expression levels" is not consistent with the sentence in Line 137 "t-SynGAP protein in mouse cortex homogenate was reduced in Syngap1+/td and Syngap1td/td mice compared to WT controls (Figure 2C)."*Reviewer #3 (Recommendations for the authors):*

SYNGAP1 is one of the most prominent risk genes for a number of neurodevelopmental disorders, including autism, intellectual disability, epilepsy and schizophrenia. However, its biological roles are still being uncovered. One aspect of this complexity is the role of its many splice variants. In this manuscript, the authors have created and analyzed several genetically modified mouse lines in which they altered the expression levels, and protein interaction sites in a subset of the isoforms of the Syngap1 protein. They then performed an extensive analysis of the mRNA and protein levels of all isoforms, behavioral and physiological phenotypes in mice, assessed seizure susceptibility, and examined cellular and molecular alterations in synapses in these mutant mice. They found that reduced expression of the α forms was damaging, while their increased expression was protective in mice.

This work is likely to be very impactful on the field, because it makes an important step forward in clarifying the roles of specific splice variants in physiological and behavioral processes relevant for neurodevelopmental disorders. One important implication of this work could be that increasing the expression of the α form using genetic tools, such as gene therapy, could have beneficial therapeutic effects in human patients.

The experiments were designed and performed at the highest standards of quality. Data were interpreted carefully. The conclusions of the study are convincing. I only have a few suggestions that can help further facilitate the interpretation of the current data:

- While Synap1 is expressed mainly in excitatory neurons, it would be useful if the authors included some single-cell transcriptomic data from the literature and public databases (like Allen Brain Atlas) and discussion on the relative distribution of Syngap1 in neuronal cell types in the cortex.

- Syngap1 is a GAP for both Ras and Rap, GTPases which have opposite effects on dendrites and synapses. The authors assessed Ras activity through p-ERK/ERK Western blots. It would be useful if they could also blot with P-BRaf/BRaf to assess Rap activity. Knowing their relative activity could help further interpret the data.

- I wonder if high mortality in Td/Td mice is due to spontaneous seizures.

- It would help to include a table summarizing all mutant mice and phenotypes.

---

## [Author Response]

[Editors’ note: the authors resubmitted a revised version of the paper for consideration. What follows is the authors’ response to the first round of review.]

Reviewer #1:This study reports the role of the PBM (PDZ-binding motif) of SynGAP in the regulation of synaptic structure, signaling, and transmission. Moreover, this study shows that the SynGAP PBM is important for setting the threshold for the dispersal of SynGAP induced by NMDAR activation. In addition, the deletion of PBM suppresses STP and LTP and impairs spatial learning, contextual fear memory, spontaneous alteration without affecting anxiety-related behavior.SynGAP is known to interact with PSD-95 in the postsynaptic density (PSD), but the function of this interaction has remained unclear. This study addresses this question using a mouse genetic approach and various in vitro and in vivo experiments. In addition, the results are of high quality and largely convincing. I think this is an important contribution to the understanding of SynGAP functions as well as the roles of the PBMs and their interactions with synaptic PDZ proteins.

Thank you for the positive comments on the impact and quality of the original manuscript.

1. The deletion of the SynGAP PBM leads to suppressed STP and LTP. I am wondering whether LTD and mGluR-LTD are also impacted because LTP and LTD are known to interfere with each other, and altered synaptic signaling might also affect LTD and mGluR-LTD, which may also affect neuronal functions.

The role of SynGAP on electrically induced LTP versus LTD has been largely resolved. Many labs, including ours, find that SynGAP is required for input specific LTP but not LTD (the one exception is mGLUR-dependent LTD in hippocampal synapses, though this is chemically induced). This assertion is best illustrated by one of our recent papers demonstrating that spike-timing-dependent plasticity in cortical pyramidal neurons from Syngap1 mice is biased toward LTD no matter the sequence of pre- post- activity (Llamosas et al., PNAS, 2021).

2. The roles of the SynGAP PBM on unitary functions of excitatory synapses remain largely unclear, which can easily be addressed by measuring mEPSCs at mutant hippocampal synapses. The increased spine diameter and surface GluA1 levels in Figure 2A suggests this possibility.

Unitary synapse analysis was performed in all three mouse lines – see new figure 9.

Reviewer #2:In this study, Kilinc et al. investigate the function of the SynGAP1 PDZ-binding motif, present in the α1 isoform, by characterizing a novel mouse model with PBM inactivating mutations. As expected, the PBM mutations in SynGAP1 abolish its interaction with PSD-95, whilst maintaining wild-type levels of the protein. Neurons from Syngap1PBM/PBM mice have reduced SynGAP1 content in dendritic spines and in PSD fractions and exhibit a range of synaptic and excitability phenotypes strikingly reminiscent of neurons from Syngap1 knockout mice. Interestingly, many of the synaptic phenotypes are rescued by NMDA receptor blockade, suggesting that NMDAR-dependent signalling causes part of the dysfunction in these neurons. In addition to altering PSD composition, neurons with the mutant SynGAP1 isoforms have altered LTP and are proposed to have a reduced threshold for NMDAR-dependent signalling. Together these phenotypes cause deficits in behaviours affecting learning and memory, similar to full Syngap1 knockouts, revealing the importance of the PBM in cognitive function.This manuscript is very well written and logically presented, addressing fundamental questions about SYNGAP1, a critically important gene in human neurodevelopment disorders and synaptic structure and function. However, despite the clear synaptic and behavioural phenotypes in the mouse model, it is still somewhat unclear how dysfunction of the α1 isoform causes these disturbances. If further experiments can solidify the proposed this mechanism, I would highly recommend this work for publication in eLife.1) One of the central posits in this manuscript is that mutation of the PBM causes increased Syngap1 mobility and lowers the threshold for NMDA receptor-dependent signalling. However, the evidence for this appears to be insufficient, relying mostly on a single experiment (Figure 3D-J). Further work to bolster these results would be highly valuable, given that discovering the function of the PBM is one of central the aims of this paper.

The new focus of the manuscript is to determine if and how endogenous SynGAP isoforms contribute to synaptic and behavioral impairments rather than how the PBM specifically regulates SynGAP functions. The new focus is now possible given the existence of two new mouse lines that inversely regulate α and β isoforms. We find that endogenous α isoforms bidirectionally regulate synapse function and behavior, and loss of their expression can account for many phenotypes present in Syngap1 mouse models. Indeed, genetically increasing expression of α isoforms can rescue intermediate phenotypes expressed in Syngap1 heterozygous mice.

2) It is difficult to appreciate how important the α1 isoform is without knowing how abundant it is compared to other isoforms in the brain. Can the authors show the relative expression of each isoform at the mRNA and/or protein level?

This has been addressed in two recent manuscripts (Araki et al., 2020; Gou et al. 2020), which suggest that each isoform is roughly equivalent in the brain – 1/3 each of alpha1/2, β, γ isoforms (though there are some differences in abundance during postnatal development). Our data support the view that these isoforms are relatively equally expressed. For example, genetic switch of β for α isoforms (B* mouse line – no change in total SynGAP protein) results in 20-30% increase in abundance of A1/A2 isoforms (Figures 3 and 4).

3) The mobility of the SynGAP1 protein appears to be a critical to its function. However, a comprehensive discussion on why SynGAP1 localization is altered after NMDAR activation, and how defects in this process leads to the observed synaptic phenotypes is lacking in the discussion. In addition, how does the movement of SynGAP1 out of the PSD affect downstream GTPase signalling? What is the proposed mechanism for how the blockade of NMDA receptors rescues the localization and synaptic phenotypes in neurons from Syngap1PBM/PBM mice? All of these themes are brought up in the paper but are not sufficiently discussed and detailed.

This has been addressed in two recent manuscripts (Araki et al., 2020; Gou et al. 2020), which suggest that each isoform is roughly equivalent in the brain – 1/3 each of alpha1/2, β, γ isoforms (though there are some differences in abundance during postnatal development). Our data support the view that these isoforms are relatively equally expressed. For example, genetic switch of β for α isoforms (B* mouse line – no change in total SynGAP protein) results in 20-30% increase in abundance of A1/A2 isoforms (Figures 3 and 4).

Reviewer #3:This manuscript by Kilinc et al. reports the plasticity-related function of SynGAP C-terminal domain in SynGAP alpha1 isoform by generating a genetically modified mouse line containing critical point mutations in the exon 21b, which disrupt the interaction between SynGAP alpha1 isoform and PSD-95. The specific alterations in the C-terminal domain induced impairment of LTP in the hippocampus and also memory-related tasks. Experiments in cultured neurons provided additional cell biological insights into the regulation of synaptic plasticity by the SynGAP alpha1 C-terminal domain. This reviewer positively evaluates the authors' findings that link the specific splice variant of SynGAP to synaptic function and memory-related circuit function. However, several points should be corrected and improved in a future manuscript.

Thank you for the positive statement.

1. The authors found increased sensitivity to weak glycine-dependent NMDAR activation in PBM/PBM neurons, which resulted in the phosphorylation of GluA1. This finding is seemingly inconsistent with the slice electrophysiology data of reduced short-term and long-term potentiation. To link these two experiments, it will be helpful to check the response of acute slices to weak stimulation (weak stimulation can induce more responses in PBM/PBM slices?) and also evaluate the level of SynGAP in PSD fraction after strong glycine-dependent NMDAR activation (less GluA1 phosphorylation in PBM/PBM neurons in culture?).

Based on the availably of the two new mouse lines and the recent publication of the Araki, Huganir et al. 2020 paper in *eLife*, the focus of the manuscript has moved away from “mobility of the isoforms” and toward how endogenous expression of these isoforms contributes to Syngap1 phenotypes. As a result, these chemical LTP studies on cultured neurons are no longer present in the revised study.

2. The underlying hypothesis of using strong and weak cLTP induction is that the difference in glycine concentration induces a different level of increase in AMPAR surface expression in culture. The validity of this hypothesis is not tested in this manuscript. Another possibility is that lower glycine concentration induces different types of plasticity, such as LTD, which is known to be easier to induce in cultured neurons. Additional information about the level of surface GluA1 or its phosphorylation between two glycine concentrations will support the hypothesis.

See point #1.

3. The basic synaptic transmission is enhanced in the perforant path-dentate gyrus granule cell synapses (Figure 2B). Still, the responses to the burst responses in the Schaffer-collateral-CA1 pyramidal neurons synapses were similar between genotypes (Figure 4A). It is desirable to show the input-output relationship in the Schaffer-collateral-CA1 pyramidal neurons synapses to test the presence of any alterations in the synaptic efficacy at the basal level.

In the revised study, basal synapse efficacy was directly addressed in all three mouse lines from the same type of neurons (L2/3 somatosensory neurons) from ex vivo slices. Please see figure 9 in the revised manuscript.

4. The authors propose that the "mobility" of the SynGAP-PBM mutant is enhanced, but no direct evidence for the alterations in the mobility is presented. It is more neutral to describe the result as "a reduced affinity" to the PSD. Otherwise, additional experiments for the measurement of protein mobility will be required.

We agree. We have adopted this suggested terminology in the revised manuscript where appropriate.

5. In Figure Supplement 1, the authors evaluated the specificity of the 06-900 antibody to the alpha1 isoform. A similar evaluation of two additional antibodies for alpha2 and β used in Figure 3 will be informative.

We are not sure as to how inclusion of this experiment would change any of our interpretations of expression within the PBM mouse. The A2 and Β antibodies have been shown to be selective for their respective C-terminal motifs. Moreover, total SynGAP expression in homogenates was not changed in the PBM mouse line.

6. In some cases, the difference between wild type and PBM/PBM mice is moderate, with p-values more than 0.01 (Figure 2F, H, Figure 3F). It is advisable to perform a power analysis to estimate the adequate sample size required for these experiments. It is also informative to show the confidence interval instead of the p-value. (See eLife 2019;8:e48175 and eNeuro. 2019 Aug 1;6(4).)

We appreciate this concept and thank you for bringing it up. Many of these experiments dealt with mobility of different isoforms in response to chemical LTP. These experiments were removed from the revised study given the new focus on endogenous expression of Α versus Β isoforms on Syngap1-related phenotypes in animals.

[Editors’ note: what follows is the authors’ response to the second round of review.]

Essential revisions:1. One of the key messages of this manuscript is the amount of α isoforms, but not the total amount of SynGAP, is critical in multiple behavioral phenotypes in mutant mice. In order to discuss the importance of isoforms, essential information is a relative abundance of each isoform in both wild-type and mutant mice. Throughout this manuscript, the comparisons within isoforms were performed, but no information about relative abundance between isoforms was provided. It is also unclear there are any differences in the relative abundance of isoforms in different brain regions studied in this manuscript, including the forebrain, hippocampus, and somatosensory cortex. The estimate of relative abundance should be provided, or any technical difficulty in this estimation should be explained.

We thank the reviewers for this comment. To account for differences in epitope to antibody affinity, true absolute quantification of different isoform would require mass spectrometry-based approach such as SureQuant. A recent publication (Araki *et al.*, 2020) carefully assessed expression of SynGAP isoforms in multiple brain regions in wild-type mice. Using custom antibodies, the authors probed for individual isoforms across development and estimated that SynGAP-α1 constitutes 35.0 ± 0.9% of total SynGAP in adult forebrain (SynGAP-α2, 44.9 ± 1.5%; SynGAP-β, 15.7 ± 0.8%; SynGAP-γ, 4.3 ± 0.3%).

Our internal data is consistent with these published findings, which also suggest that SynGAP-α1 and SynGAP-α2 are the most abundant in adult forebrain. In Figure 2, we show that reduction in α1 and α2 coincides with a reduction in total SynGAP despite a large increase observed for SynGAP-β in +/Td mice. Similarly, in Figure 3 we observe an increasing trend for total SynGAP which coincides with an increase in α1 and α2 signal while SynGAP-β expression is eliminated in β*/β* mouse.

The published estimates of C-terminal isoform abundance are included in the revised introduction.

2. If alpha1 is the only isoform expressed in the brain, the interpretation of introducing PSD95-nonbinding mutant is straightforward, but in the presence of other isoforms, dominant-negative effects of PBM mutant should be considered. It is possible that PBM mutant complexed with different isoforms impair localization or function of other isoforms, leading to an effect similar to the reduction of total SynGAP protein. Clarification of this point by additional experimental data is necessary.

In the revised Results section for Figure 6, we include new data that addresses the concern of heterodimerization of SynGAP isoforms and possible dominant negative effects of a1 isoforms with PBM mutations. These new data can be found in two places: (1) new panels in Figure 6; (2) a new supplemental figure (Figure 6 – supplement).

First, we include additional panels in Figure 6 that include isoform-specific expression in the PSDs of WT and PBM mice. These new data demonstrate that there is no change in relative abundance of a2 and b isoforms in the PSDs from whole hippocampus taken from PBM mice. The original figure only included PSD levels using a pan-SynGAP antibody. The original data indicated that there was less total SynGAP (using a pan-isoform antibody) in the PSD from PBM mouse hippocampi. The newly included panels extend these results by including isoform abundance data (revised Figure 6A – two additional panels). These new data indicate that the reduction in total SynGAP PSD levels is driven by reduced a1 levels because a2 and b abundance did not change in these same samples (and trended toward an increase). While we did not blot for g isoforms, it is very low expressed relative to the others (see point 1 above), indicating that changes to this isoform would not be sufficient to drive the reduction in total SynGAP from PSDs. We did not blot for a1 directly in these experiments because the mutated PBM reduces affinity for a1 antibodies. These data suggest that a1 may act independently from the other C-terminal variants in the PSD.

Experiments in Supplemental Figure 6 were designed to more directly address the possibility that mutated a1 in PBM mice influenced other isoforms in the PSD, perhaps through hetero-multimerization of SynGAP isoforms. This experiment was also designed to mitigate reduced a1-antibody affinity caused by PBM mutations. We performed weak and strong chemical LTP (cLTP) paradigms in cultured primary neurons from WT and PBM mice. Strong cLTP drives extrusion of total SynGAP (measured with pan-antibody) from the PSD(Figure S6A), which accompanies increases in synapse signaling and insertion of AMPARs (Araki et al., 2020). However, we found that weak cLTP does not extrude SynGAP (Figure S6B), which is consistent with past studies (Araki et al., 2020). In contrast, weak cLTP stimulations were capable of driving reduction in total SynGAP from PSDs in PBM mice. Blotting with isoform-specific antibodies in the weak cLTP studies provided insight into the differential behavior of total SynGAP in PSDs from WT mice compared to PBM mice. To arrive at these measurements, we compared relative expression of SynGAP proteins in the PSD in control and weak cLTP conditions within different wells of the same cultured neuron plates (either WT or PBM mice). Thus, the reduced affinity of a1 antibodies in this experiment is not an issue. In WT neurons, weak cLTP was sufficient to drive reduced PSD abundance for both a2 and b isoforms. However, a1 expression was unchanged, demonstrating that this isoform exhibits distinct behavior within biochemical fractions in response to synaptic NMDAR activation. Replicating this approach in PBM neurons revealed that the distinct biochemical behavior of a1 relative to the other isoforms was due to the existence of an intact PBM motif. Indeed, disrupting the PBM resulted in a1 behaving similar to the other isoforms during weak cLTP. As a result, these data suggest that the reduction in total SynGAP in WT PSDs that occurs in response to strong cLTP can be explained through dynamics of a1. A reasonable biochemical explanation for this difference can be attributed to decreased potential mobility of a1 homomers due to increased basal affinity for binding partners with PDZ domains. Presumably, this reduced basal affinity for binding partners in the PSD is lost because of PBM mutagenesis. Thus, PBM mutagenesis removes a unique biochemical property a1 isoforms, which has functional consequences for neuronal function and behavior (Figures 6, 8, and 9). Moreover, the significant reduction in a2 and b in WT PSDs after weak cLTP does not drive changes in total SynGAP levels likely because a1 is substantially enriched in the PSD relative to the other isoforms (Gou et al., 2020).

Together, these data add substantially to our understanding of how different SynGAP isoforms behave in the PSD in response to NMDAR signaling. This is evidence supporting the idea that SynGAP C-terminal isoforms can behave independently of each within the PSD in response to NMDAR activation. These new data suggest that mutant a1 is not acting as a dominant negative through sequestering the other isoforms. Rather, it suggests that a1 is largely in complex with itself within the PSD and responds differently to NMDAR activation compared to the other isoforms. Further work will be required to fully establish the nature and composition of SynGAP isoform complexes and how they respond to biological stimuli.

3. Figure 9A-C. The sample size (n = 8) for mEPSC measurements in the PBM/PBM mice seems small to draw a safe conclusion, and thus should be increased, or the authors should explicitly state the limits associated with interpretation of this sample size.

Collectively, our data demonstrate that PBM mutations drive a form of *Syngap1* loss-of-function. Analysis of *m*EPSCs from various cortical and hippocampal excitatory neurons with reduced *Syngap1* gene expression demonstrate increased excitatory synapse strength, particularly in development. Thus, increased excitatory synapse strength in developing cortical neurons with a disrupted a1 PBM is generally consistent with *Syngap1* loss of function. N=8 is on the low side for mEPSC analysis but not critically low, and the results obtained in this experiment in PBM mice (Figure 9) are consistent with increased AMPAR dendritic spine surface expression and reduced synapse and PSD expression of SynGAP protein in neurons from the same mouse line (Figure 6). In the revised Results section for Figure 9, we include a cautionary note that the sample size for this ePhys experiment is on the low side.

4. It is unclear why the authors selected Morris water maze as the second behavioral test, which is not designed to evaluate long-lasting memory. The rationale behind selecting two distinct memory-related behavioral tests, remote contextual memory test and water maze-based training performance, should be explained.

We included these two paradigms in order to assess learning and memory mechanisms known to be sensitive to *Syngap1* gene function. MWM learning results from plasticity in the hippocampus and entorhinal cortex driven by new associations connecting location of the hidden platforms to visual cues surrounding the maze. Learning can be evaluated through assessment of the acquisition curves. Memory can be independently evaluated from learning through inclusion of a probe test. *Syngap1* heterozygous null mice exhibit deficits in MWM acquisition (i.e., poor spatial learning), which has been affirmed in three past studies from three different groups using three distinct strains of *Syngap1* heterozygous null mice (Komiyama et al., 2002; Kim et al., 2003; Muhia et al., 2010). In each of these past studies, there was no impact of *Syngap1* heterozygosity on the probe test, demonstrating no “spatial memory deficits” in this strain. Thus, MWM acquisition (i.e., spatial learning) is highly sensitive to *Syngap1* expression, which means that this paradigm can be used to assess how *Syngap1* gene function impacts hippocampal/cortical-associated learning mechanisms.

In contrast, remote contextual memory (RCM) in *Syngap1* mice is a test of long-term memory rather than learning. This paradigm drives a learned association between the training context and an aversive footshock. Retrieval of the memory is tested 30 days after training – the “remote period”. During this 30-day period, information originally encoded in the hippocampus and amygdala is slowly integrated into cortical circuits for long-term storage. Past studies in *Syngap1* heterozygous null mice have repeatedly shown that there is no phenotype 1 day after training (*regular long-term memory*; Guo et al., 2008; Muhia et al., 2010; Creson et al., 2019), but phenotypes emerge selectively when memory is retrieved 30 days later (*remote memory*; Creson et al., 2019). Thus, this task can reveal how *Syngap1* gene function contributes to long-term memory mechanisms resulting from systems consolidation mechanisms. As a result, by including these two behavioral paradigms, we were able to broaden our survey of how individual isoforms impact both learning and memory mechanisms.

5. The phenotype in the Morris water maze is difficult to interpret. SynGAP heterozygote mice show impairment, but IRES-TD mice showed no change. This is inconsistent with other behavioral tests. Β* mice showed improvement in the training session, but the 24 h probe test did not show a difference. It is difficult to connect the data with the result of contextual fear memory. This point should be discussed.

IRES-TD mice run in the MWM were heterozygous for the targeted allele. Thus, they only had ~50% reduction of a1/2 isoforms and ~0-25% reduction in total SynGAP protein (Figures 2 and 2S). This drove some phenotypes but others were unaffected (such as MWM). This suggests that MWM is sensitive to loss of total SynGAP protein. We could not test Td/Td mice because they died shortly after weaning, but we suspect based on the totality of all the data in the three lines (and *Syngap1* het nulls), MWM would be affected in Td/Td mice if we were able to test them. b* mice do show improved acquisition rates but no change in the probe test, which is consistent with improved (faster acquisition of) spatial learning but normal memory. We hypothesize that this may be due to increase in a1/a2 expression in these mice. It is conceivable that improvements in spatial learning are sensitive modest increases in α isoform expression, even though modest decreases in the same isoforms do not drive impairments. This is quite an exciting result because a1 expression could serve as a platform for understanding molecular and cellular mechanisms that improve cognitive functions in the forebrain. Indeed, we are currently planning a new project that will directly test this hypothesis – we are creating a series of conditional, inducible SynGAP-isoform mice that can be crossed to *Syngap1* Het null mice. These planned studies are beyond the scope of the current study.

Regarding RCM, as mentioned above in Point #4, this paradigm drives unique forms of learning and memory and therefore may have different sensitivities to *Syngap1* isoform expression relative to MWM. As a result, it is reasonable for distinct *Syngap1* lines to have different learning/memory phenotypes. These differences suggest that SynGAP isoforms function in different cell types across the forebrain to support various neural mechanisms for learning/memory.

Aspects of Point#5 are now included in the revised Results section for Figure 2.

6. While Synap1 is expressed mainly in excitatory neurons, it would be useful if the authors included some single-cell transcriptomic data from the literature and public databases (like Allen Brain Atlas) and discussion on the relative distribution of Syngap1 in neuronal cell types in the cortex.

Thank you for this comment. We spent considerable time mining the Allen Brain single cell sequencing data for *Syngap1*. These efforts led to a new figure (Figure 10) in the revised manuscript. In the revised manuscript, we discussed these new data in the context of prior studies that addressed SynGAP expression/function in excitatory versus inhibitory neurons. Briefly, we found that *Syngap1* expression is several-fold higher, on average, in glutamatergic versus GABAeric neurons in cortex + hippocampus mouse data sets. There was even less transcript detected in non-neuronal cells. Within the glutamatergic populations, the highest expression was in more superficial cortical neurons (L5a, L4, L2/3 IT neurons). There was slightly less expression in large output channel neurons, such as L5 PT neurons and CA1 neurons. These scRNA-seq data agree with experimental evidence of SynGAP protein expression in rodent neurons. Roger Tsien’s group reported that SynGAP protein expression was enriched in upper lamina of cortex and this pattern was consistent with high protein levels in excitatory synapses. Mary Kennedy’s group demonstrated that SynGAP protein was absent in several types of GABAergic neurons, but was expressed in a subpopulation of morphologically distinct inhibitory neurons. The scRNA-seq data also agree with prior experimental observations from electrophysiological and behavioral measurements in mice. Commonly observed and robust phenotypes observed in conventional *Syngap1* null mice were phenocopied in mice where *Syngap1* null heterozygosity was restricted to excitatory neurons in the forebrain (EMX1 population; Ozkan et al., 2014). Moreover, major electrophysiological and behavioral phenotypes in *Syngap1* heterozygous mice were rescued when *gene* expression was restored in in the EMX1 population (Ozkan et al., 2014). In contrast, our group observed minor phenotypes in mice when *Syngap1* expression was disrupted in GABAergic neurons (GAD2 population; Ozkan et al., 2014). Another group found similar results. In that study, most *Syngap1* Heterozygous null mouse phenotypes were insensitive to selective disruption in a GABAergic neuron population, though one behavioral measure of cognition was mildly affected (Berryer et al., 2016).

7. Syngap1 is a GAP for both Ras and Rap GTPases, which have opposite effects on dendrites and synapses. The authors assessed Ras activity through p-ERK/ERK Western blots. It would be useful if they also blot with P-BRaf/BRaf to assess Rap activity. Knowing their relative activity would help further interpret the data.

We appreciate the comment and agree that these experiments would be enlightening. However, we believe that this level of analysis is beyond the scope of the present study. We have ongoing projects in the lab that are exploring potential dichotomy of SynGAP function on synapses and dendritic morphogenesis, with the hypothesis that SynGAP may regulate distinct GTPases within sub-cellular compartments to drive differential effects on synapses relative to dendrites. This study extends to different types of glutamatergic neurons (distinct lamina and brain areas in vivo). The potential differences in SynGAP function across distinct sub-cellular regions could be enabled through alternatively spliced sequences. As a result, we do plan on performing these experiments, but they would be part of new study to be disclosed in a future manuscript.

8. A table with all mutants and phenotypes could help the readers.

At the suggestion of the reviewers, we have created a comprehensive table (Table 1) that presents protein expression, ePhys, and behavioral results for all lines tested.